# Regional study of mode-2 internal solitary waves in the Pacific coast of Central America using marine seismic survey data

Wenhao Fan[1], Haibin Song[1], Yi Gong[1], Shun Yang[1], Kun Zhang[1]

[1] State Key laboratory of Marine Geology, School of Ocean and Earth Science, Tongji University, Shanghai, 200092, China

*Correspondence to*: Haibin Song (hbsong@tongji.edu.cn)

**Abstract.** In this paper, a regional study of the mode-2 internal solitary waves (ISWs) in the Pacific coast of Central America is carried out by using the seismic reflection method. The observed relationship between the dimensionless propagation speed and the dimensionless amplitude (DA) of the mode-2 ISW is analyzed. When DA<1.18, the dimensionless propagation speed seems to increase with the increasing dimensionless amplitude, divided into two parts with different growth rates. When DA>1.18, the dimensionless propagation speed increases with the increasing dimensionless amplitude at a relatively small growth rate. We suggest that the influences of seawater depth (submarine topography), pycnocline depth, and pycnocline thickness on the propagation speed of the mode-2 ISW in the study area, cause the relationship between the dimensionless propagation speed and the dimensionless amplitude to diversify. The observed relationship between the dimensionless wavelength and the dimensionless amplitude of the mode-2 ISW is also analyzed. When DA<1, the nondimensional wavelengths seem to change from 2.5 to 7 for a fixed nondimensional amplitude. When DA>1.87, the dimensionless wavelength increases with the increasing dimensionless amplitude. Additionally, the seawater depth has a great influence on the wavelength of the mode-2 ISW in the study area. Overall the wavelength increases with the increasing seawater depth. As for the vertical structure of the amplitude of the mode-2 ISW in the study area, we find that it is affected by the nonlinearity of the ISW and the pycnocline deviation (especially the downward pycnocline deviation).

## 1 Introduction

The amplitude and propagation speed of the mode-1 ISW are larger than those of the mode-2 ISW. The mode-1 ISWs are more common in the ocean. In recent years, with the advancement of observation instruments, the mode-2 ISWs in the ocean have been gradually observed, such as on the New Jersey shelf (Shroyer et al., 2010), in the South China Sea (Liu et al., 2013; Ramp et al., 2015; Yang et al., 2009), at Georges Bank (Bogucki et al., 2005), over Mascarene Ridge in the Indian Ocean (Da Silva et al., 2011)), and on the Australian North West Shelf (Rayson et al., 2019). Conventional physical oceanography observation and remote sensing observation have their limitations. That is, the horizontal resolution of conventional physical oceanography observation methods (such as mooring) is low. And satellite remote sensing cannot see the ocean interior. Seismic oceanography (Holbrook et al., 2003; Ruddick et al., 2009; Song et al., 2021), as a new oceanography survey method, has a high spatial resolution (the vertical resolution and horizontal resolution can reach about 10m). It can better describe the

spatial structure and related characteristics of mesoscale and small scale phenomena in the ocean (Biescas et al., 2008, 2010;
Fer et al., 2010; Holbrook & Fer, 2005; Holbrook et al., 2013; Pinheiro et al., 2010; Sallares et al., 2016; Sheen et al., 2009;
Tsuji et al., 2005). Scholars have used the seismic oceanography method to carry out related studies on the geometry and
kinematics characteristics (mainly related to propagation speed) of ISW in the South China Sea, the Mediterranean Sea, and
the Pacific Coast of Central America (Bai et al., 2017; Fan et al., 2021a, 2021b; Geng et al., 2019; Sun et al., 2019; Tang et
al., 2014, 2018).
At present, the researches on the mode-2 ISW are mainly based on simulation. Through simulation, scholars have found
that the pycnocline deviation will affect the stability of the mode-2 ISW. And it will make the top and bottom structure of the
mode-2 ISW asymmetrical (Carr et al., 2015; Cheng et al., 2018; Olsthoorn et al., 2013). The instability caused by the
pycnocline deviation mainly appears at the bottom of the mode-2 ISW. It is manifested in that the amplitude of the mode-2
ISW peak is smaller than the amplitude of the trough. Because the upper sea layer is thinner than the bottom sea layer. The
wave tail will appear similar to K-H instability billow and the wave core will appear small-scale flip (Carr et al., 2015; Cheng
et al., 2018). For the propagation speed of the mode-2 ISW, scholars found through simulation experiments that it increases
with the increasing amplitude (Maxworthy, 1983; Salloum et al ., 2012; Stamp & Jacka, 1995; Terez & Knio, 1998). Brandt
et al. (2014) simulated the material transport of mode-2 ISW with large amplitude in the laboratory. They found that when
$2a/h_2>4$ ($a$ is the amplitude of the mode-2 ISW and $h_2$ is the pycnocline thickness, we define the dimensionless amplitude
$\tilde{a}=2a/h_2$ for the convenience of using in the following text), the linear relationship between the propagation speed (wavelength)
and the amplitude is destroyed. That is, when the amplitude $\tilde{a}\geq4$, the propagation speed increases relatively slowly, and the
wavelength increases rapidly. They believe that the above results are caused by strong internal circulation related to the very
large amplitude and the influence of the top and bottom boundaries. Chen et al. (2014) calculated the KdV propagation speed
and the fully nonlinear propagation speed of the ISW as the function of the pycnocline depth and the pycnocline thickness,
respectively. They found the propagation speed of the mode-2 ISW increases monotonously with the increasing pycnocline
depth, and firstly increases and then decreases with the increasing pycnocline thickness. Carr et al. (2015) found by simulations
that the pycnocline deviation has little effect on the propagation speed, wavelength, and amplitude of the mode-2 ISW.
Maderich et al. (2015) found that for the mode-2 ISWs when the dimensionless amplitude $\tilde{a}<1$, the deep-water weakly
nonlinear theory (Benjamin, 1967) can describe the numerical simulation and experimental simulation results well. When $\tilde{a}>1$,
the wavelength (propagation speed) increases with the amplitude faster than the results predicted by the deep-water weakly
nonlinear theory. But the solution of Kozlov and Makarov (1990) can well estimate the corresponding wavelength and
propagation speed when the amplitude is $1<\tilde{a}<5$. Terletska et al. (2016) found that the propagation speed and amplitude of the
mode-2 ISW will decrease after passing the step. And the closer the mode-2 ISW is to the step in the vertical direction at the
time of incidence, the smaller the propagation speed and amplitude of the mode-2 ISW are after passing the step. Kurkina et
al. (2017) used GDEM (Generalized Digital Environmental Model) to find that the seawater depth in the South China Sea is
the main controlling factor of the mode-2 ISW propagation speed. And the propagation speed increases exponentially with the
increasing seawater depth. Deepwell et al. (2019) found by simulation that the relationship curve that the mode-2 ISW
propagation speed increases with the increasing amplitude has a strong quadratic fitting relationship. They speculated that this
quadratic fitting relationship comes from the influence of seawater depth (when the seawater depth is smaller, the propagation
speed is also smaller).
The simulation can well reveal the kinematics characteristics of the mode-2 ISW. But the actual ocean conditions are
often more complicated, which is manifested by the diversity of controlling factors in the kinematics process. The observations
including the seismic oceanography method are also required to continually provide a basic understanding of the geometry
and kinematics characteristics of the mode-2 ISW. For example, limited by factors such as the lower spatial resolution of the
observation methods, previous scholars have less direct observation research on the propagation speed and wavelength of the
mode-2 ISW in the ocean. And there is even less research (including observation research) on the vertical structure of the
mode-2 ISW. The seismic oceanography method has more advantages for carrying out the above-mentioned research due to
its higher spatial resolution. The Pacific coast of Central America (western Nicaragua) has relatively continuous submarine
topography along the coastline, including the continental shelf and continental slope, with a seawater depth of 100-2000m (Fig.
1). At present, there is relatively little research work on internal waves in this area. We reprocessed the historical seismic data
in this area and identified a large number of mode-2 ISWs with relatively complete spatial structures in the region. This
discovery is very helpful to carry out observation research on the geometry and kinematics characteristics of the mode-2 ISW.
Fan et al. (2021a, 2021b) used the multichannel seismic data of the survey lines L88 and L76 (cruise EW0412, see Fig. 1 for
the survey line locations) in the Pacific coast of Central America to respectively report the mode-2 ISWs in this area and study
the shoaling features of the mode-2 ISW in this area. However, a single survey line can only reveal the local characteristics of
the mode-2 ISW in the study area. A deep understanding of the geometry and kinematics characteristics (mainly related to
propagation speed) of the mode-2 ISW in the study area requires a regional systematic study. In this work, we reprocessed the
seismic data of the entire study area. And we identified numerous mode-2 ISWs on multiple survey lines in the region (the
positions of the observed ISWs and the survey lines they located are shown by the black filled circles and the red lines in Fig.
1, respectively). Based on the numerous mode-2 ISWs observed by multiple survey lines in the study area, this paper will
conduct a regional study on the characteristic parameters. These characteristic parameters include the pycnocline deviation
degree, propagation speed, and wavelength of the mode-2 ISW, as well as the vertical structure characteristics of the mode-2
ISW amplitude in the study area.

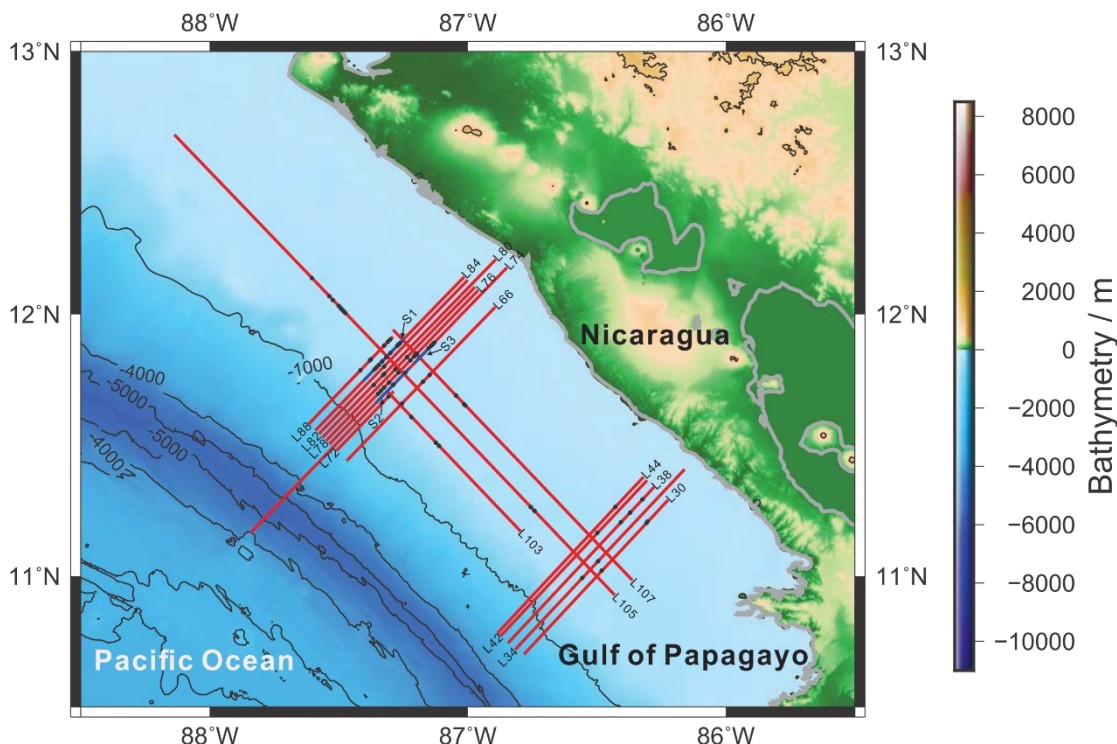

**Figure 1. Distribution of multichannel seismic data. The red lines indicate the positions of the survey lines, and the black filled circles on the red lines indicate the positions of the observed mode-2 ISWs. The blue lines S1, S2, and S3 are part of the seismic sections containing the mode-2 ISWs, which will be displayed in Fig. 3 and Fig. 4.**

## 2 Data and Methods

This paper mainly uses seismic reflection to conduct a regional study on the mode-2 ISWs in the Pacific coast of Central America. The seismic data of the cruise EW0412 used in this study was provided by the MGDS (The Marine Geoscience Data System) (http://www.marine-geo.org/). The cruise EW0412 collected high-resolution multichannel seismic data from the continental shelf to the continental slope in the coastal areas of Sandino Forearc Basin, Costa Rica, Nicaragua, Honduras, and El Salvador (Fulthorpe & McIntosh, 2014). The seismic acquisition parameters of the cruise EW0412 are as follows: the sampling interval is 1 ms, each shot gather has 168 traces, the shot interval is 12.5 m, the trace interval is 12.5 m, and the minimum offset is 16.65 m. The seawater seismic reflection sections used in this study were obtained through the following processing processes: defining geometry, noise attenuation, common midpoint (CMP) sorting, velocity analysis, normal moveout (NMO), stacking, and post-stack denoising. Previous scholars demonstrated that seismic reflections generally track isopycnal surfaces (Holbrok et al., 2013; Krahmann et al., 2009; Sheen et al., 2011). We believe the seismic stacked sections (like Figs. 3 and 4) include the information of the density profile. So we do not provide the plots of the density profile the waves propagate on (even in schematic form) in the following sections.

108      In this research, we try to use the maximum amplitude (the maximum vertical displacement of isopycnals) to study the amplitude related characteristics of mode-2 ISW, like the relationship between the propagation speed and the maximum amplitude in Fig. 9. But the correlativity is not very strong. We also noticed that the amplitude, defined as the maximum vertical displacement of isopycnals, is used less in quantitatively describing the amplitude related characteristics of mode-2 ISW. Particularly, in mode-2 ISW simulation research, the scholars often use the dimensionless amplitude $\tilde{a}$ to quantitatively describe the amplitude-related characteristics of mode-2 ISW, like the relationship between the propagation speed and the dimensionless amplitude (Brandt et al., 2014; Carr et al., 2015). It is important to point that in mode-2 ISW simulation research, the dimensionless amplitude the scholars used comes from the three-layer model. But the mode-2 ISW in the actual ocean has a multilayer structure (multiple continuous density displacements above and below the mid-depth of the pycnocline). It is different from the three-layer model used in the simulation experiment to describe the convex mode-2 ISW. As for the three-layer model, the upper layer of the convex mode-2 ISW is the peak and the lower layer is the trough. Because there is almost no work of the previous scholars to define the dimensionless amplitude of the mode-2 ISW based on the mode-2 ISW in the actual ocean (with multiple continuous density displacements above and below the mid-depth of the pycnocline) for our reference. To compare our observation results to the simulation results and quantitatively describe the amplitude-related characteristics of mode-2 ISW, we try our best to build an equivalent three-layer model. The equivalent three-layer model results from the mode-2 ISW with the continuous structure in the actual ocean. It should be pointed that the equivalent three-layer model is defined by trying our best to analogize with the three-layer model. Therefore it is not completely the same as the three-layer model like Fig. 1 in Brandt et al. (2014). We use the equivalent three-layer model to calculate the equivalent amplitude, the equivalent pycnocline thickness, and the equivalent wavelength of the mode-2 ISW. Similarly, as the equivalent three-layer model is defined by trying our best to analogize with the three-layer model, the equivalent amplitude (dimensionless amplitude) is not completely equivalent to the one used by Brandt et al. (2014). For the mode-2 ISW with a multilayer structure, the sum of all ISW peak amplitudes $a_p$ and the sum of all ISW trough amplitudes $a_t$ are respectively taken as the equivalent peak and trough amplitude of the mode-2 ISW with a three-layer model structure. Then the equivalent amplitude of the mode-2 ISW with a three-layer model structure is the average of $a_p$ and $a_t$. And the equivalent pycnocline thickness is calculated by $h_2=h-a_p-a_t$, where $h$ is the seawater thickness affected by the mode-2 ISW with a multilayer structure. The equivalent wavelength of the mode-2 ISW with a three-layer model structure is the average of all ISW peak and trough wavelengths in the multilayer structure. The detailed calculation process is described in Fan et al. (2021a). This study uses an improved ISW apparent propagation speed calculation method to calculate the apparent propagation speed of ISW. This method firstly does pre-stack migration of the common offset gather sections. And then picking the CMP and shot point pairs corresponding to the ISW trough or peak from the pre-stack migration sections of different offsets with a high signal-to-noise ratio. By fitting the CMP-shot point pairs, we can calculate the apparent propagation speed and apparent propagation direction of the ISW. The ISW horizontal velocity can be expressed by the equation as follow:

$$v = \frac{cmp2-cmp1}{T} = \frac{cmp2-cmp1}{(s2-s1)dt} \tag{1}$$

where *cmp1* and *cmp2* are the peak or trough position of the ISW at different time, *s1* and *s2* are the shot numbers corresponding
to *cmp1* and *cmp2*, and *dt* is the time interval of shots. The detailed calculation process is described in Fan et al. (2021a).
The wavelength of the mode-2 ISW is usually defined as half-width at half-amplitude of the ISW (Carr et al., 2015; Stamp
& Jacka, 1995), as shown by $\lambda$ in Fig. 2. In a seismic survey, the sound is sent from a towed source, reflected from aquatic
structures, and received by an array of towed hydrophones with time delays that depends on the geometry of the ray paths
taken. The detailed introduction to seismic principles is described by Ruddick et al. (2009). Traditional seismic reflection
imaging assumes that the underground structure is fixed. Since the mode-2 ISWs move relatively fast in the horizontal direction
(about 0.5m/s) during the seismic acquisition process, the seismic reflection imaging of the mode-2 ISWs needs to consider
the influence of the horizontal motion of the ISW. The wavelength of the mode-2 ISW observed by the seismic reflection
method is the apparent wavelength. The apparent wavelength of the mode-2 ISW is controlled by the relative motion direction
of the ship and the ISW, the ship speed, and the propagation speed of the ISW. The propagation speed of the mode-2 ISW
(about 0.5m/s) is generally lower than the ship speed (about 2.5m/s) during seismic acquisition. When correcting the apparent
wavelength of the mode-2 ISW to obtain the actual wavelength, it is divided into two situations in which the motion direction
of the ISW and the ship is the same and opposite, as shown in Fig. 2. When the ISW and the ship move in the same direction,
the wavelength (apparent wavelength) estimated from the seismic stacked section is larger. That is, the wavelength (apparent
wavelength $\lambda_s$) of the ISW observed on the seismic stacked section denoted by the blue curve in Fig. 2a is greater than the
wavelength $\lambda$ of the actual ISW at the beginning and end respectively denoted by the black and red curves in Fig. 2a. The
influence of the horizontal movement of the ISW should be eliminated When correcting the apparent wavelength $\lambda_s$ to obtain
the actual wavelength $\lambda$, it is necessary to subtract the distance $x_w$ moved by the ISW within the seismic acquisition time
corresponding to the apparent wavelength distance of the ISW. That is:

$$\lambda = \lambda_s - x_w = \lambda_s - \frac{\lambda_s}{V_{ship}} V_{water} \tag{2}$$

where $V_{ship}$ is the ship speed, and $V_{water}$ is the propagation speed of the ISW (Fig. 2a).
When the ISW and the ship move in the opposite direction, the wavelength (apparent wavelength) estimated from the
seismic stacked section is smaller. That is, the wavelength (apparent wavelength $\lambda_s$) of the ISW observed on the seismic stacked
section denoted by the blue curve in Fig. 2b is smaller than the wavelength $\lambda$ of the actual ISW at the beginning and end
respectively denoted by the black and red curves in Fig. 2b. The influence of the horizontal movement of the ISW should be
eliminated When correcting the apparent wavelength $\lambda_s$ to obtain the actual wavelength $\lambda$, it is necessary to add the distance $x_w$
moved by the ISW within the seismic acquisition time corresponding to the apparent wavelength distance of the ISW. That is:

$$\lambda = \lambda_s + x_w = \lambda_s + \frac{\lambda_s}{V_{ship}} V_{water} \tag{3}$$

where $V_{ship}$ is the ship speed, and $V_{water}$ is the propagation speed of the ISW (Fig. 2b).

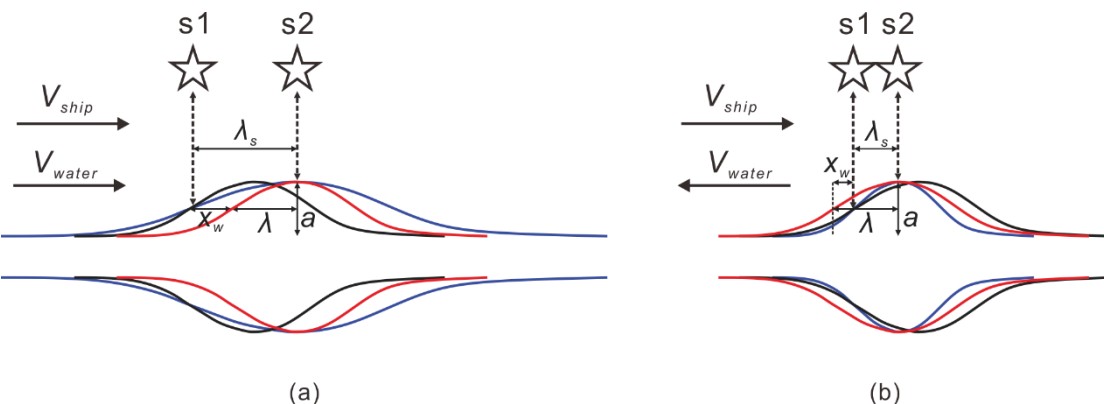


**Figure 2. Schematic diagram of the apparent wavelength correction of the mode-2 ISW. (a) The ISW moves in the same direction as**
**the ship. (b) The ISW moves in the opposite direction to the ship. S1 denotes the self-excitation and self-reception position of the ship**
**at 1/2 amplitude of the ISW at the beginning. S2 denotes the self-excitation and self-reception position of the ship at the peak of the**
**amplitude of the ISW. $V_{ship}$ is the ship speed, and $V_{water}$ is the ISW propagation speed. $\lambda_s$ is the apparent wavelength of the ISW**
**observed by the seismic stacked section. $\lambda$ is the actual wavelength of the ISW. $a$ is the amplitude of the ISW. $x_w$ is the distance moved**
**by the ISW during the time the ship moves from S1 to S2. The black curve denotes the ISW at the beginning. The red curve denotes**
**the ISW moved $x_w$ distance from the starting position. And the blue curve denotes the ISW observed on the seismic stacked section.**
**3 Results and Interpretations**
**3.1 Typical Sections Interpretation and Regional Distribution Characteristics of the Mode-2 ISWs**

182       In addition to the survey lines L88 and L76 with mode-2 ISWs observed by Fan et al. (2021a, 2021b), we also found

mode-2 ISWs on many other survey lines in the study area. Two typical survey lines are L84 and L74 (see the red lines in Fig.
1 for the locations of these two survey lines). Figure 3 shows the partial seismic stacked section S1 of the survey line L84 (see
the blue line in Fig. 1 for the location of this section S1). We have identified 10 mode-2 ISWs from the seismic section S1 (see
Fig. 3 for their positions and corresponding numbers. ISW1-ISW4 are located at the shelf break, and ISW5-ISW10 are located
on the continental shelf). And calculated the characteristic parameters of these 10 mode-2 ISWs, such as the seafloor depth
(seawater depth) $H$, maximum amplitude (in the vertical direction), equivalent amplitude $a$, equivalent pycnocline thickness
$h_2$, dimensionless amplitude $\tilde{a}$, mid-depths of the pycnocline $h_c$ , the degree to which the mid-depth of the pycnocline deviates
from 1/2 seafloor depth $P_d$, equivalent wavelength $\lambda$, dimensionless wavelength (we define the dimensionless wavelength
$\lambda_0=2\lambda/h_2$ for the convenience of using in the following text), and apparent propagation speed $U_c$ (Table 1). The equivalent
wavelength and the dimensionless wavelength in Table 1 have been corrected using Eq. (2) (the ISWs have the same motion
direction as the ship, and the ISWs with the large propagation speed estimation error have been corrected using a propagation
speed of 0.5 m/s). The maximum amplitudes of the ISWs ISW1-ISW7 on the survey line L84 are all less than 10 m. And the
maximum amplitudes of ISW8-ISW10 are larger, around 15 m. The $\tilde{a}$ values of these ten mode-2 ISWs on the survey line L84
are all less than 2 (Table 1). We define the ISW, whose $\tilde{a}$ value is less than 2, as the mode-2 ISW with a small amplitude. And
define the ISW, whose $\tilde{a}$ value is larger than 2, as the mode-2 ISW with a large amplitude. The ten mode-2 ISWs on the survey
line L84 belong to the mode-2 ISWs with small amplitude. The $\tilde{a}$ values of ISW8, ISW9, and ISW10 are around 1. Their
amplitudes are slightly larger in these small-amplitude mode-2 ISWs. When calculating the $P_d$ values, it is found that except
for the pycnocline centers of ISW8, ISW9, and ISW10 are deeper than 1/2 seafloor depths, the pycnocline centers of the other
seven mode-2 ISWs are shallower than 1/2 seafloor depths (Table 1). For ISW1, ISW2, and ISW3, the $P_d$ values are both
greater than 20%, which appear as the asymmetry of waveforms (the asymmetry of the front and rear waveform, and the
asymmetry of the top and bottom waveform). When the $P_d$ value is small, the waveform of the mode-2 ISW is more
symmetrical, such as ISW8, ISW9, and ISW10. The waveforms of ISW1, ISW2, and ISW3 at the shelf break are asymmetrical.
And their dimensionless wavelengths $\lambda_0$ are significantly larger than the $\lambda_0$ values of the ISWs on the continental shelf which
have the same level of dimensionless amplitudes ($\tilde{a}$) (for example, the $\tilde{a}$ value of ISW2 is 0.45, and the value of $\lambda_0$ is 9.55; the
value of $\tilde{a}$ of ISW7 is 0.42, and the value of $\lambda_0$ is 3.49). It makes the overall relationship between dimensionless wavelength $\lambda_0$
and the dimensionless amplitude $\tilde{a}$ are not absolute linear correlation (the $\lambda_0$ increases with the increasing $\tilde{a}$). The apparent
propagation speeds $U_c$ of the 10 mode-2 ISWs on the survey line L84 are about 0.5 m/s. And the apparent propagation
directions are all shoreward. For ISWs with small apparent propagation speed calculation errors in shallow water (ISW6, ISW7,
and ISW9), the $U_c$ does not strictly increase with the increasing $\tilde{a}$. For example, the $\tilde{a}$ value of ISW6 is 0.4, and the $U_c$ value
is about 0.58 m/s. The $\tilde{a}$ value of ISW9 is 1.19, and the $U_c$ value is about 0.38 m/s.

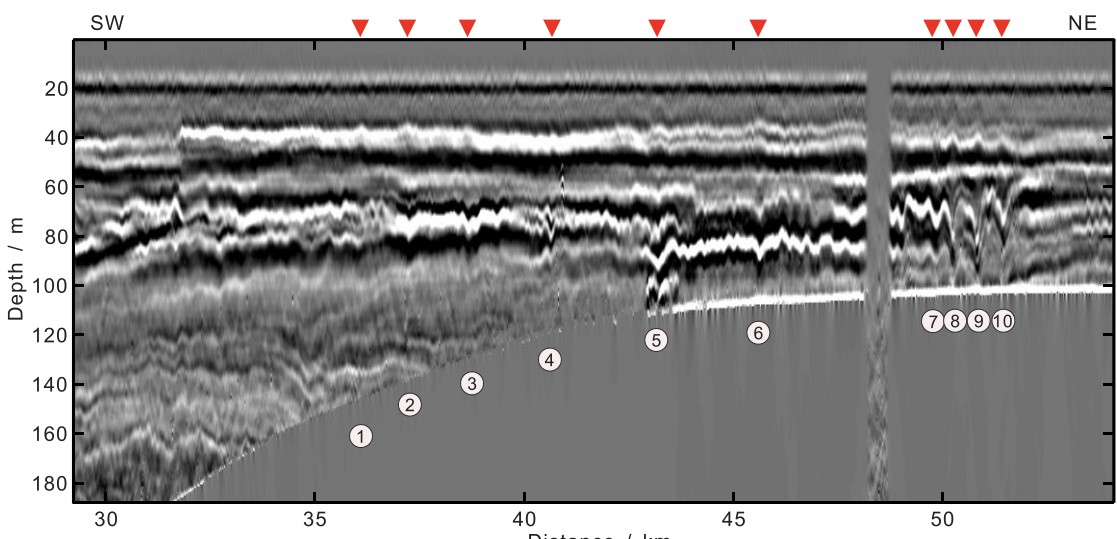

**Figure 3. Seismic stacked section S1, observed mode-2 ISWs part on the survey line L84. Arrows and numbers indicate the ten**
**identified mode-2 ISWs ISW1-ISW10. The location of the S1 seismic stacked section is shown in Fig. 1. The horizontal axis indicates**
**the distance to the starting point of the survey line L84. The survey line L84 acquisition time is from 07:15:14 on 17 December 2004,**
**to 17:26:49 on 17 December 2004.**

**Table 1. Characteristic Parameters of the 10 Mode-2 Internal Solitary Waves in Survey Line L84.**

| ISW# | $H$ | $A$ | $a$ | $h_2$ | $2a/h_2$ | $h_c$ | $P_d$ | $\lambda$ | $2\lambda/h_2$ | $U_c$ | $\alpha$ | $C$ |
|------|-----|-----|-----|-------|----------|-------|-------|-----------|----------------|-------|----------|-----|

| | (m) | (m) | (m) | (m) | | (m) | (%H) | (m) | | (m/s) | (s⁻¹) | (m/s) |
|---|---|---|---|---|---|---|---|---|---|---|---|---|
| ISW1 | 145.5 | 3 | 2.22 | 29.23 | 0.15 | 54.88 | 24.6 | 103.6 | 7.09 | 0.85±0.6 | -0.018 | 0.384 |
| ISW2 | 138.8 | 4.7 | 5.84 | 25.93 | 0.45 | 51.31 | 26.1 | 123.8 | 9.55 | 0.69±0.19 | -0.0179 | 0.382 |
| ISW3 | 130.5 | 4.1 | 4.45 | 27.6 | 0.32 | 49.05 | 24.8 | 84.6 | 6.13 | 0.52±0.12 | -0.0181 | 0.378 |
| ISW4 | 121.5 | 5.2 | 6.04 | 34.72 | 0.35 | 59.4 | 2.2 | 55.18 | 3.18 | 0.19±0.11 | -0.018 | 0.372 |
| ISW5 | 111 | 6.79 | 12.67 | 40.84 | 0.62 | 51.31 | 7.6 | 95.38 | 4.67 | 0.32±0.16 | 0.0068 | 0.391 |
| ISW6 | 108 | 4.6 | 7.5 | 37.19 | 0.4 | 48.48 | 10.2 | 50.61 | 2.72 | 0.58±0.16 | 0.0108 | 0.389 |
| ISW7 | 104.3 | 6.4 | 7.34 | 34.83 | 0.42 | 48.11 | 7.8 | 60.86 | 3.49 | 0.64±0.28 | 0.0158 | 0.386 |
| ISW8 | 103.5 | 13.2 | 15.82 | 32.94 | 0.96 | 53.38 | -3.2 | 72.97 | 4.43 | 0.46±0.24 | 0.0155 | 0.385 |
| ISW9 | 103.5 | 15.9 | 13.56 | 22.79 | 1.19 | 52.81 | -2.1 | 88.47 | 7.76 | 0.38±0.17 | 0.0161 | 0.385 |
| ISW10 | 102.8 | 13.6 | 15.87 | 20.62 | 1.54 | 52.62 | -2.4 | 94.1 | 9.13 | 0.55±0.34 | 0.0164 | 0.384 |

Note. $H$, seafloor depths. $A$, maximum amplitudes. $a$, equivalent ISW amplitudes. $h_2$, equivalent pycnocline thicknesses. $h_c$, the mid-depths of the pycnocline. $P_d$, the degree to which the mid-depth of the pycnocline deviates from 1/2 seafloor depth. $\lambda$, equivalent wavelengths. $U_c$, apparent propagation speeds obtained from seismic observation. $\alpha$, quadratic nonlinear coefficient shown in Ea. (9) and is obtained by solving Eq. (6). $C$, linear phase speed which is obtained by solving Eq. (6).

The survey line L74 is located in the southeast direction of the survey line L84 (see Fig. 1 for the specific location). Figure 4 shows the partial seismic stacked sections (S2 and S3) of the survey line L74. We have identified seven mode-2 ISWs from the seismic sections S2 and S3. Their positions and corresponding numbers are shown in Fig. 4. And the statistical characteristic parameters are shown in Table 2. The equivalent wavelength and the dimensionless wavelength in Table 2 have been corrected using Eq. (2) (the ISWs have the same motion direction as the ship, and the ISWs with the large propagation speed estimation error have been corrected using a propagation speed of 0.5 m/s). The maximum amplitudes of the ISWs ISW12-ISW17 on the survey line L74 are all less than 10 m. And the maximum amplitude of ISW11 is larger, 13.6 m. The $\tilde{a}$ values of these seven mode-2 ISWs are all less than 2 (Table 2). They are the mode-2 ISWs with small amplitude. And the amplitude of ISW11 is slightly larger among them. When calculating the $P_d$ value, it is found that the pycnocline centers of the mode-2 ISWs ISW11-ISW17 are all deeper than 1/2 of the seafloor depths (Table 2). Except for ISW11 (the bottom reflection event is broken), as for the other six mode-2 ISWs ISW12-ISW17, the $P_d$ values are both greater than 15%. The asymmetry of ISW12 and ISW13 is manifested in that the connection between the top peaks of the ISW and the bottom troughs of the ISW is not vertical. The pycnocline center of ISW14 deviates from 1/2 of the seafloor depth the most, which is 51.5%. Its asymmetry is manifested in the large difference between the top and bottom waveforms near the pycnocline center. ISW15, ISW16, and ISW17 are located on the continental shelf, and their pycnocline deviations are larger. But their waveforms are more symmetrical than other ISWs. When the downward pycnocline deviation is large, the influence of pycnocline deviation on the stability of the mode-2 ISW is more complicated than when the pycnocline deviates upwards. And it may be controlled

by factors such as wavelength. There is no absolute linear correlation relationship between the dimensionless wavelengths $\lambda_0$
and the dimensionless amplitudes $\tilde{a}$ of the seven mode-2 ISWs on the survey line L74 (the $\lambda_0$ increases with the increasing $\tilde{a}$).
For example, the $\tilde{a}$ values of ISW12 and ISW14 are greater than that of ISW16. But the $\lambda_0$ value of ISW16 is greater than the
$\lambda_0$ values of ISW12 and ISW14. The apparent propagation speeds $U_c$ of the seven mode-2 ISWs on the survey line L74 are
about 0.5 m/s. And their propagation directions are all shoreward. For the ISWs in shallow water whose apparent propagation
speed calculation errors are small (ISW12, ISW14, ISW16, and ISW17), the $U_c$ value generally increases with the increasing
$\tilde{a}$.

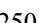


**Figure 4. (a) and (b) are respectively the seismic stacked sections S2 and S3, observed mode-2 ISWs parts on the survey line L74.**
**The arrows and the numbers indicate the seven identified mode-2 ISWs ISW11-ISW17. The locations of the seismic stack section S2**
**and S3 are shown in Fig. 1. And the horizontal axis indicates the distance to the starting point of the survey line L74. The survey line**
**74 acquisition time is from 06:31:03 on 3 December 2004 to 02:30:01 on 4 December 2004.**

**Table 2. Characteristic Parameters of the Seven Mode-2 Internal Solitary Waves in Survey Line L74.**

| ISW# | $H$ (m) | $A$ (m) | $a$ (m) | $h_2$ (m) | $2a/h_2$ | $h_c$ (m) | $P_d$ (%H) | $\lambda$ (m) | $2\lambda/h_2$ | $U_c$ (m/s) |
|------|---------|---------|---------|-----------|----------|-----------|------------|---------------|----------------|-------------|
| ISW11 | 138.8 | 13.6 | 24.19 | 26.98 | 1.79 | 73.38 | -5.7 | 83.05 | 6.16 | 0.19±0.1 |
| ISW12 | 103.5 | 7.31 | 9.95 | 32.82 | 0.61 | 60.34 | -16.6 | 68.11 | 4.15 | 0.63±0.08 |
| ISW13 | 90.75 | 5.68 | 6.08 | 36.22 | 0.34 | 55.62 | -22.6 | 94.41 | 5.21 | 0.49±0.24 |
| ISW14 | 92.25 | 6.86 | 11.17 | 35.04 | 0.64 | 68.35 | -51.5 | 50.69 | 2.89 | 0.49±0.21 |
| ISW15 | 90 | 5.46 | 8.91 | 38.94 | 0.46 | 52.1 | -15.8 | 112.7 | 5.79 | 0.36±0.26 |
| ISW16 | 91.5 | 5.74 | 8.67 | 39.53 | 0.44 | 57.97 | -26.7 | 100.7 | 5.09 | 0.60±0.17 |
| ISW17 | 91.5 | 6.4 | 12.71 | 32.56 | 0.78 | 57.6 | -25.9 | 69.56 | 4.27 | 1.07±0.2 |

**Note. $H$, seafloor depths. $A$, maximum amplitudes. $a$, equivalent ISW amplitudes. $h_2$, equivalent pycnocline thicknesses. $h_c$, the mid-**
**depths of the pycnocline. $P_d$, the degree to which the mid-depth of the pycnocline deviates from 1/2 seafloor depth. $\lambda$, equivalent**
**wavelengths. $U_c$, apparent propagation speeds obtained from seismic observation.**

In addition to the survey lines L74 and L84, the mode-2 ISWs also have sporadic distribution on other survey lines in the
area (see the black filled circles in Fig. 1). We have identified 70 mode-2 ISWs in the study area. They appeared from 2
December 2004 to 18 December 2004. On 17 December 2004 and 18 December 2004, there were more mode-2 ISWs (Fig.
5a), 21 (10 for survey line L84, 6 for survey line L88, and 5 for survey line L76) and 9 (1 for survey line L72, 5 for survey line
L76, and 3 for survey line L103) respectively. Observe the distribution of the appearance time of mode-2 ISWs observed in
the study area in Fig. 5a (in days). It can be found that the mode-2 ISWs frequently appeared on the northwest side of the study
area in December 2004, and appeared in early and late December. In addition, the spatial distribution range of the mode-2
ISWs is large, ranging from the continental slope to the continental shelf (see Figs. 1, 3, and 4). Figure 5b shows the time when
the mode-2 ISWs observed in the study area appeared in hours. Combined with Fig. 5a, it can be found that from 2 December
2004 to 8 December 2004, the ISWs appeared at around 12:00 and 00:00 (or 24:00) in a day. From 10 December 2004 to 13
December 2004, the ISWs appeared at around 12:00 and 24:00 in a day, and relatively more appeared around 12:00. From 14
December 2004 to 18 December 2004, the ISWs appeared at around 12:00 and 00:00 (or 24:00) in a day, and relatively more
appeared around 00:00 (or 24:00). The survey lines L103, L105, and L107 are perpendicular to the propagation direction of
the mode-2 ISWs in the study area (Fig. 1). Therefore, these three survey lines are not included in the subsequent statistical
analysis of the mode-2 ISW characteristic parameters. We have counted the characteristic parameters of 53 mode-2 ISWs in
the study area. In these 53 mode-2 ISWs, there are 51 small-amplitude ISWs ($\tilde{a}<2$). And there are 40 ISWs with smaller
amplitude ($\tilde{a}<1$) among these 51 small-amplitude ISWs (Fig. 6a). The mode-2 ISWs in the study area are dominated by smaller
amplitudes (Fig. 6a). The maximum amplitudes (in the vertical direction) of the mode-2 ISWs mainly change in the range of
3-13 m (Fig. 6d). And the equivalent wavelengths of most of the mode-2 ISWs are on the order of about 100 m (Fig. 6c, the
equivalent wavelength in the figure has been corrected according to Eq. (2) and Eq. (3)). When calculating the propagation
speed of the mode-2 ISW, due to the low signal-to-noise ratio of some survey lines, the calculation errors of some ISWs
propagation speeds are relatively large. Therefore, when analyzing the apparent propagation speed of the mode-2 ISW of the
study area, we only used 26 ISWs with relatively small errors (the error is less than half of the calculated value). The apparent
propagation speeds of the mode-2 ISWs in the study area are on the order of 0.5 m/s (Fig. 6b). And most of the mode-2 ISWs
propagate in the shoreward direction. We have traced back the time when each ISW in the study area (mainly the ISWs located
on the continental shelf) appeared at the continental shelf break using the ISW propagation speed of 0.5 m/s, as shown in Fig.
5c, in hours. Combined with Fig. 5a, it is found that from 2 December 2004 to 8 December 2004, the ISWs traced back to the
continental shelf break appeared at around 12:00 and 00:00 (or 24:00) in a day, and relatively more appeared around 12:00.
From 10 December 2004 to 13 December 2004, most of the ISWs traced back to the continental shelf break appeared at around
24:00 (or 0:00) in a day. From 14 December 2004 to 18 December 2004, the ISWs traced back to the continental shelf break
appeared at around 12:00 and 24:00 (or 0:00) in a day. The mode-2 ISWs observed in the study area may be generated by the
interaction between the internal tide and the continental shelf break.

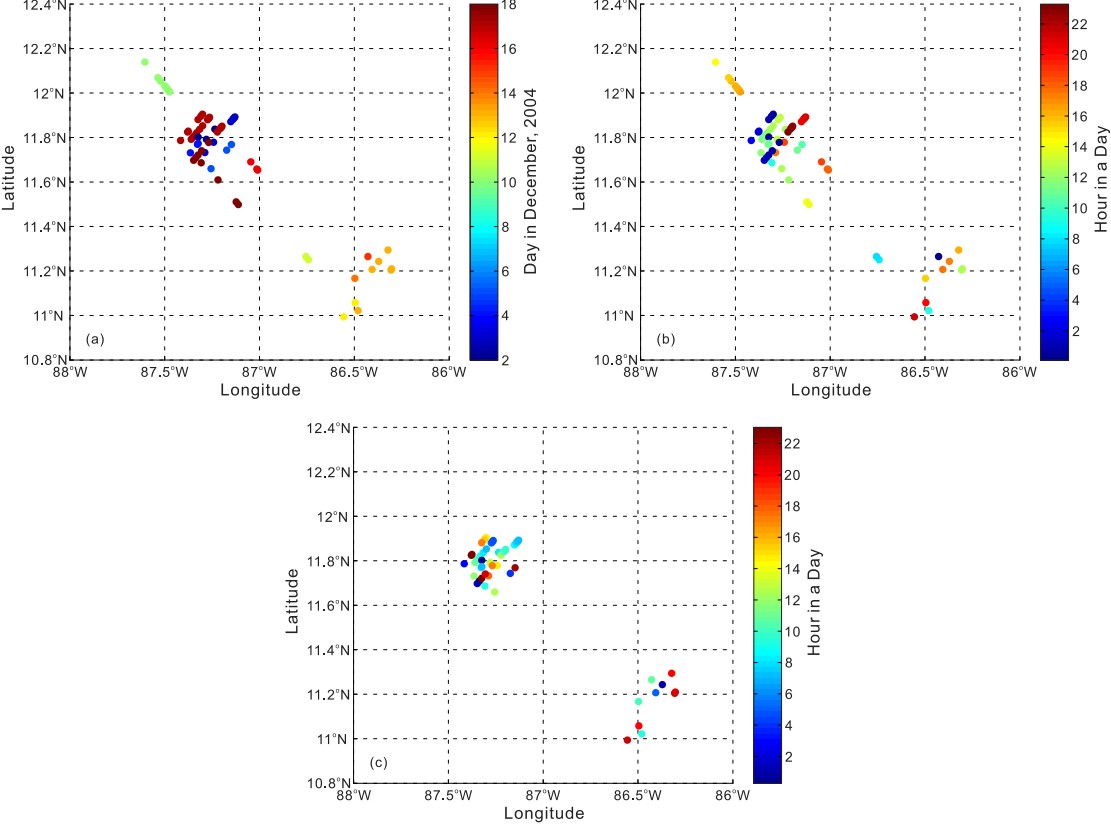


**Figure 5. (a) The time when the mode-2 ISWs observed in the study area appeared in days. (b) The time when the mode-2 ISWs observed in the study area appeared in hours. (c) Tracing back the time (in hours) when internal solitary waves appeared at the continental shelf break in the study area.**

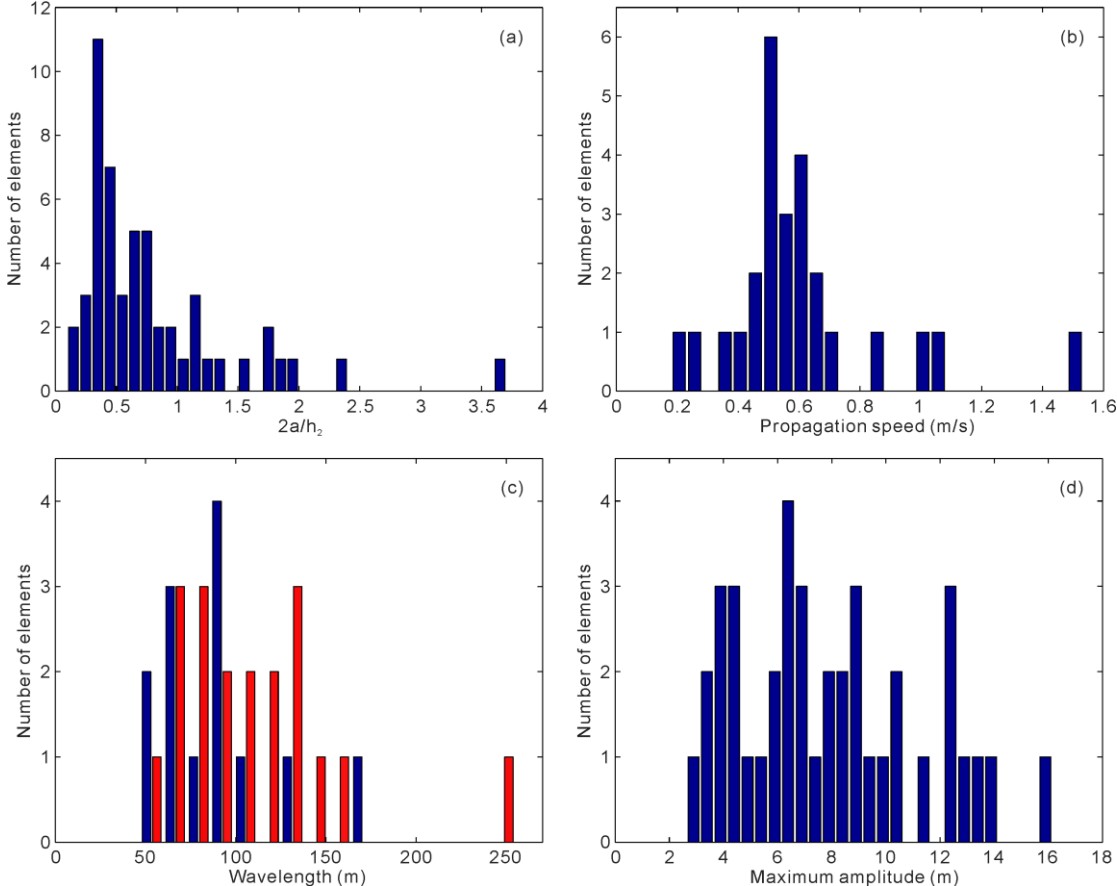

**Figure 6. (a) The histogram of the dimensionless amplitude of the mode-2 ISW in the study area. (b) The histogram of the propagation speed of the mode-2 ISW in the study area. (c) The histogram of the wavelength of the mode-2 ISW in the study area. The dark blue and red colour bars denote the ISWs on the survey lines in the SW-NE direction and in the NE-SW direction, respectively. (d) The histogram of the maximum amplitude of the mode-2 ISW in the study area.**

### 3.2 Propagation Speed and Wavelength Characteristics of the Mode-2 ISW in Study Area

Inspired by the work of Maderich et al. (2015) and Chen et al. (2014), we respectively calculated the relationships between the dimensionless propagation speed and the dimensionless amplitude $\tilde{a}$, the dimensionless wavelength $\lambda_0$ and the $\tilde{a}$, the propagation speed ($U_c$) and the maximum amplitude $A$, the wavelength ($\lambda$) and the $A$, the $U_c$ and the pycnocline depth, the $U_c$ and the pycnocline thickness. Figure 7 shows the relationship between the dimensionless propagation speeds (we define the dimensionless propagation speed $\tilde{U}=U_c/C$ for the convenience of using in the following text) and the dimensionless amplitudes

$\tilde{a}$ of the observed 26 mode-2 ISWs (with relatively small errors) in the study area. When $\tilde{a}<1.18$, it seems that the relationship
between the $\tilde{U}$ values and the $\tilde{a}$ values of the observed mode-2 ISWs in the study area has the trends respectively given by
Kozlov and Makarov (1990), as well as Salloum et al. (2012). That is, the $\tilde{U}$ of the mode-2 ISW increases with the increasing
$\tilde{a}$, but with different growth rates. The fitting effects of Kozlov and Makarov (1990), Salloum et al. (2012), and the
segmentation fitting in Fig. 7 are shown in Table 3. The segmentation fitting computed by ourselves in Fig. 7 can be expressed
by the equation as follow:
$$\tilde{U} = \frac{9.441\tilde{a}^4 - 27.19\tilde{a}^3 + 28.14\tilde{a}^2 - 10.93\tilde{a} + 1.016}{\tilde{a} - 0.6401} \tag{4}$$

When $\tilde{a}>1.18$, the relationship between the $\tilde{U}$ values and the $\tilde{a}$ values of the observed mode-2 ISWs in the study area is closer
to the result predicted by the deep-water weakly nonlinear theory (Benjamin, 1967). That is, The $\tilde{U}$ of the mode-2 ISW
increases with the increasing $\tilde{a}$ at a relatively small growth rate. The fitting effect of Benjamin (1967) in Fig. 7 is shown in
Table 3. Figure 8 shows the relationship between the dimensionless wavelengths $\lambda_0$ and the dimensionless amplitudes $\tilde{a}$ of the
observed 32 mode-2 ISWs (there are 13 ISWs on the survey lines in the SW-NE direction, and 19 ISWs on the survey lines in
the NE-SW direction, see Fig. 6c) in the study area. In Fig. 8, the black and red crosses denote the ISWs on the survey lines in
the SW-NE direction and in the NE-SW direction, respectively. The survey line in the SW-NE direction is consistent with the
movement direction of the ISWs. Use Eq. (2) to correct the apparent wavelength to obtain the actual wavelength. The survey
line in the NE-SW direction is opposite to the movement direction of the ISWs. Use Eq. (3) to correct the apparent wavelength
to obtain the actual wavelength. Figure 8 shows the result after correcting the apparent wavelength of the ISW. When using
Eq. (2) and Eq. (3) to correct the apparent wavelength, the propagation speed of the ISW estimated in Fig. 7 needs to be used.
The dimensionless wavelengths $\lambda_0$ of the ISWs with the large error in the estimation of the propagation speed are not shown
in Fig. 8. Observing Fig. 8, it can be found that when $\tilde{a}<1$, the relationship between the $\lambda_0$ values and the $\tilde{a}$ values of the
observed mode-2 ISWs in the study area is closer to the result predicted by the deep-water weakly nonlinear theory (Benjamin,
1967). But the $\lambda_0$ values change from 2.5 to 7 for a fixed $\tilde{a}$ value. The fitting effect of Benjamin (1967) in Fig. 8 is shown in
Table 4. When $\tilde{a}>1.87$, the relationship between the $\lambda_0$ values and the $\tilde{a}$ values of the observed mode-2 ISWs in the study area
is closer to the solution of Salloum et al. (2012). That is, the $\lambda_0$ of the mode-2 ISW increases with the increasing $\tilde{a}$. The fitting
effects of Salloum et al. (2012) and the segmentation fitting in Fig. 8 are shown in Table 4. The segmentation fitting computed
by ourselves in Fig. 8 can be expressed by the equation as follow:
$$\lambda_0 = 1.865\tilde{a} + 2.066 \tag{5}$$

When $1<\tilde{a}<1.87$, the $\lambda_0$ values of the observed mode-2 ISWs in the study area are higher than those predicted by the deep-
water weakly nonlinear theory (Benjamin, 1967) and Salloum et al. (2012).

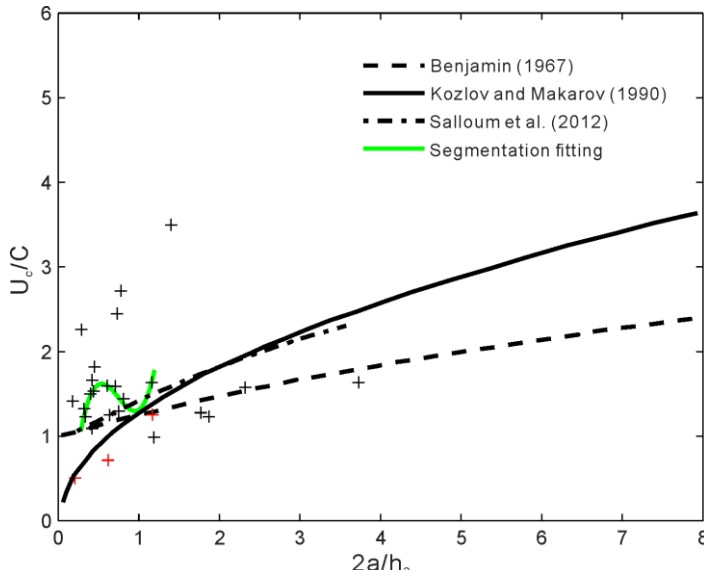


**Figure 7. The relationship between the dimensionless propagation speeds and the dimensionless amplitudes of the mode-2 ISWs**
**observed in the study area. The black crosses denote the seismic observation results of the mode-2 ISWs.**

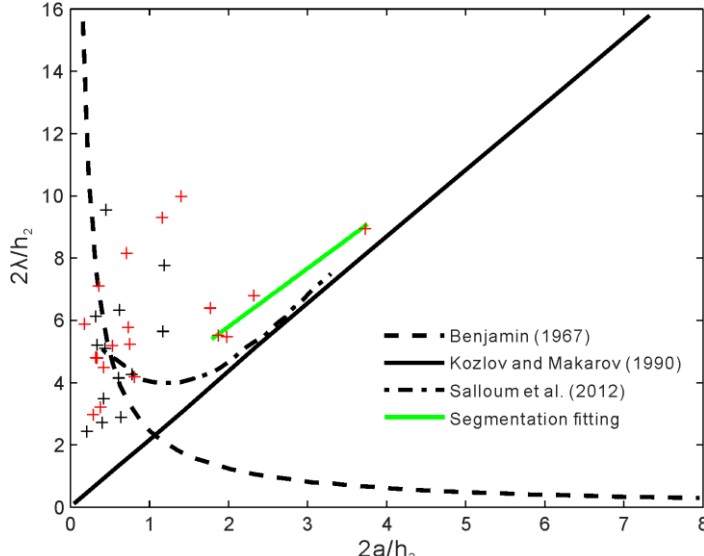


**Figure 8. The relationship between the dimensionless wavelengths and the dimensionless amplitudes of the mode-2 ISWs observed**
**in the study area. The black and red crosses denote the ISWs on the survey lines in the SW-NE direction and in the NE-SW direction,**
**respectively.**

**Table 3. The Fitting Effects of Each Curve in Figure 7 on the Observation Points.**

| $\tilde{a}$ range | fitting curve | $R^2$ |
|---|---|---|

| | | |
|---|---|---|
| larger than 1.18 | Benjamin (1967) | 0.34 |
| smaller than 1.18 | Kozlov and Makarov (1990) | 0.67 |
| smaller than 1.18 | Salloum et al. (2012) | less than 0 |
| smaller than 1.18 | segmentation fitting | 0.39 |

**Note. For the fitting curve of Kozlov and Makarov (1990), we use the three red cross observation points to compute the $R^2$ value. For the fitting curves of Salloum et al. (2012) and segmentation fitting, we use the black cross observation points, whose $\tilde{U}$ are less than 2, to compute the $R^2$ values.**

**Table 4. The Fitting Effects of Each Curve in Figure 8 on the Observation Points.**

| $\tilde{a}$ range | fitting curve | $R^2$ |
|---|---|---|
| larger than 1.87 | Salloum et al. (2012) | less than 0 |
| larger than 1.87 | segmentation fitting | 0.97 |
| smaller than 1 | Benjamin (1967) | less than 0 |

The relationship between the propagation speeds $U_c$ and the maximum amplitudes $A$ of the mode-2 ISWs observed in the study area is shown in Fig. 9a. The relationship between the wavelengths $\lambda$ and the maximum amplitudes $A$ is shown in Fig. 9b. It can be found that the $U_c$ and $\lambda$ of the mode-2 ISW in the study area are less affected by the $A$. There is no obvious linear correlation between $U_c$ and $A$, as well as between $\lambda$ and $A$ (Figs. 9a and 9b). When the $A$ values are between 6 m and 11 m, the variety range of $U_c$ is relatively large. And there is a significant increase in $U_c$ (Fig. 9a). When the $A$ values are between 7 m and 13 m, there is a significant increase in wavelength $\lambda$ (Fig. 9b). The relationship between the propagation speeds $U_c$ and the pycnocline depths $h_c$ of the observed mode-2 ISWs in the study area is shown in Fig. 10a. And the relationship between the propagation speeds $U_c$ and the pycnocline thicknesses $h_2$ is shown in Fig. 10b. As for the observed mode-2 ISWs in the study area, their $h_c$ values are mainly concentrated in the range of 40-70 m (Fig. 10a). And their $h_2$ values are mainly concentrated in the range of 10-60 m (Fig. 10b). As with the numerical simulation results of Chen et al. (2014), the $U_c$ values of the observed mode-2 ISWs in the study area seem to have the trends to increase slowly with the increasing $h_c$ values and $h_2$ values, respectively. The fitting effects of Chen et al. (2014) in Fig. 10 are shown in Table 4. The trends mentioned above are not completely monotonous in Fig. 10. It is manifested as the large variation range of the $U_c$ on the vertical axis. We analyze it is caused by the fact that other factors (such as seawater depth), other than the pycnocline depth $h_c$ and the pycnocline thickness $h_2$, also affect the propagation speed $U_c$.

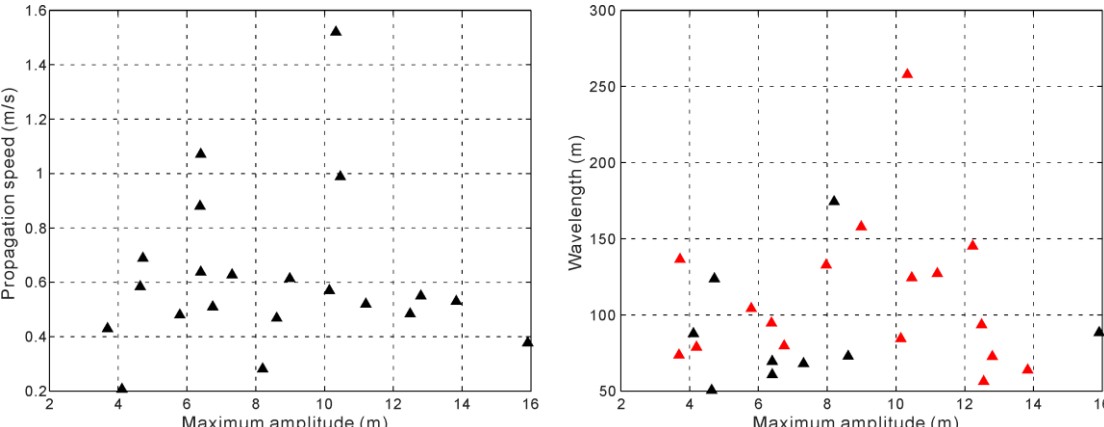


**Figure 9. (a) The relationship between the propagation speeds and the maximum amplitudes of the mode-2 ISWs observed in the study area. (b) The relationship between the wavelengths and the maximum amplitudes of the mode-2 ISWs observed in the study area. The black and red crosses denote the ISWs on the survey lines in the SW-NE direction and in the NE-SW direction, respectively.**


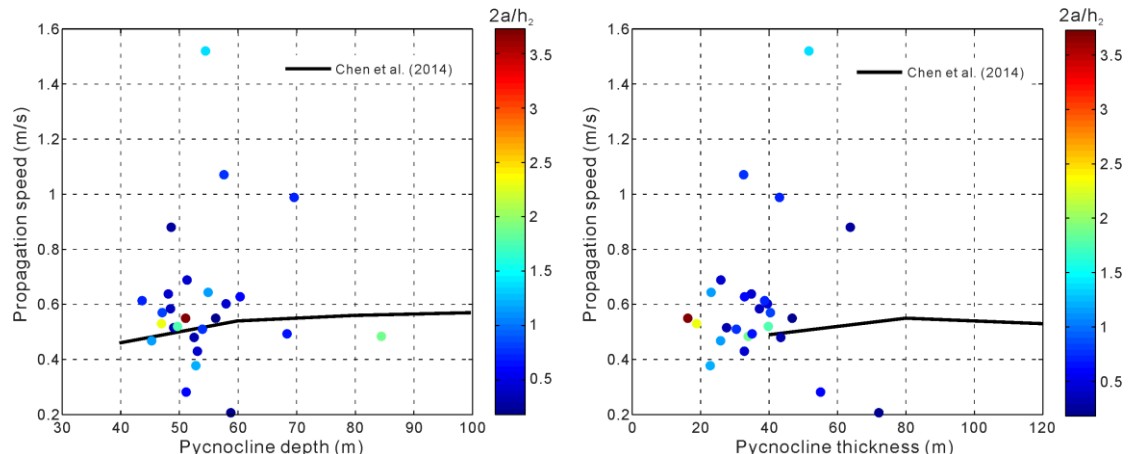


**Figure 10. (a) The relationship between the propagation speeds and the pycnocline depths of the mode-2 ISWs observed in the study area. (b) The relationship between the propagation speeds and the pycnocline thicknesses of the mode-2 ISWs observed in the study area. The color filled circle indicates the dimensionless amplitude.**


**Table 5. The Fitting Effects of Each Curve in Figure 10 on the Observation Points.**

| figure | fitting curve | $R^2$ |
|---|---|---|
| Figure 10a | Chen et al. (2014) | less than 0 |
| Figure 10b | Chen et al. (2014) | less than 0 |

**Note. For the fitting curve of Chen (2014) in Fig. 10a, we use the observation points, whose propagation speeds are less than 0.8 m/s**
**and larger than 0.21 m/s, to compute the $R^2$ value. For the fitting curve of Chen (2014) in Fig. 10b, we use the observation points,**
**whose propagation speeds are less than 0.9 m/s and pycnocline thicknesses are larger than 40 m, to compute the $R^2$ value.**

### 3.3 Vertical Structure Characteristics of the Mode-2 ISW Amplitude in Study Area

The vertical distribution of ISW amplitude (the vertical displacement of isopycnal) is called its vertical structure. ISWs have different modes, which correspond to different vertical structures (Fliegel and Hunkins, 1975). Previous scholars have used different theoretical models to study the vertical structure of ISW amplitude (Fliegel and Hunkins, 1975; Vlasenko et al., 2000; Small and Hornby, 2005). Among them, only Vlasenko et al. (2000) compared the results of numerical simulation with the results of local observations. And they found that the depths correspond to the ISW maximum amplitude (the maximum vertical displacement of isopycnals) given by the two are in good agreement with each other. At present, there is less work comparing the theoretical vertical structure of mode-2 ISW amplitude with the observed results. This work is conducive to improving our understanding of the vertical structure of the mode-2 ISW in the ocean (including the factors that affect the vertical structure). It can also test the validity and applicability of the theoretical vertical structure to a certain extent. The seismic oceanographic method has a high spatial resolution, and its clear imaging results of ISWs are more conducive to the study of vertical structure. The vertical structure of ISW amplitude is controlled by a variety of environmental factors. Geng et al. (2019) used the seismic oceanography method to study the vertical structure of ISW amplitude near Dongsha Atoll in the South China Sea. It is found that when the ISW interacts intensely with the seafloor, the observed vertical structure of ISW amplitude may be significantly different from the theoretical result. Gong et al. (2021) compared the vertical structure of ISW estimated by theoretical models with the vertical structure of ISW observed by the seismic oceanography method. And they analyzed in detail the factors affecting the vertical structure of ISW amplitude near Dongsha Atoll in the South China Sea. It is found that the vertical structure of ISW is mainly controlled by nonlinearity. It usually appears that the quadratic nonlinear coefficients of ISWs that conform to the linear vertical structure function are small, while the quadratic nonlinear coefficients of ISWs conforming to the first-order nonlinear vertical structure function are larger. In addition, topography, ISW amplitude, seawater depth, and background flow may all affect the vertical structure of ISW amplitude. It appears that larger seawater depth may weaken the influence of the nonlinearity of the ISW on the vertical structure, making the vertical structure of ISW more in line with linear theory. Larger amplitude will make ISW more susceptible to the influence of topography, which will change the vertical structure. Vlasenko et al. (2000) observed that the vertical structure of ISW has local extrema. They thought it is caused by smaller-scale internal waves. In addition, the background flow shear also has an important effect on the vertical structure (Stastna et al., 2002; Liao et al., 2014). Xu et al. (2020) found that the background flow at the center of the eddy can weaken the amplitude of ISW.

Observing the vertical structure of the mode-2 ISW amplitude in the study area, it is found that they follow the following characteristics as a whole. The amplitude of ISW in the upper half of the pycnocline decreases with the increasing seawater depth. The amplitude of ISW in the lower half of the pycnocline firstly increases and then decreases with the increasing seawater depth (see Figs. 11 and 12 in this paper, Fig. 5 of Fan et al., 2021a, and Fig. 6 of Fan et al., 2021b). Due to the influence of the pycnocline center deviation on the development of the vertical structure of the ISW amplitude, the vertical

structure of the mode-2 ISW amplitude in the study area generally only exhibits part of the characteristics given by the vertical
mode function. As for the vertical mode function, the amplitude of the ISW in the upper and lower half of the pycnocline
firstly increases and then decreases with the increasing seawater depth, respectively, as shown by the blue and red curves in
Figs. 11 and 12. Since the pycnocline centers of most of the mode-2 ISWs observed in the study area deviate upwards, the
ISW structure at the top is not as well developed as the ISW structure at the bottom. Therefore, the amplitude of ISW in the
upper half of the pycnocline usually decreases with the increasing seawater depth. Figure 11 shows the vertical structures of
the amplitude of the 10 mode-2 ISWs ISW1-ISW10 in the survey line L84. The pycnocline centers corresponding to ISW1-
ISW7 all deviate upwards (see the degree to which the mid-depth of the pycnocline deviates from 1/2 seafloor depth in Table
1, the positive sign indicates that the pycnocline deviates upward, and the negative sign indicates that the pycnocline deviates
downward). Among them, ISW1-ISW4 (Fig. 11a-d) and ISW7 (Fig. 11g) were only picked up one reflection event in the upper
half of the pycnocline. From ISW6 (Fig. 11f), it can be seen that the amplitude of ISWs in the upper half of the pycnocline
decreases with the increasing seawater depth. From ISW2 (Fig. 11b), ISW4 (Fig. 11d), ISW5 (Fig. 11e), and ISW7 (Fig. 11g),
it can be seen that the amplitude of ISW in the lower half of the pycnocline firstly increases and then decreases with the
increasing seawater depth. The pycnocline centers corresponding to ISW8-ISW10 all slightly deviate downwards (see the
degree to which the mid-depth of the pycnocline deviates from 1/2 seafloor depth in Table 1, the positive sign indicates that
the pycnocline deviates upward, and the negative sign indicates that the pycnocline deviates downward). From ISW8 (Fig.
11h) and ISW10 (Fig. 11j), it can be seen that the amplitude of ISW in the upper half of the pycnocline decreases with the
increasing seawater depth. Figure 12 shows the vertical structures of the amplitude of the four mode-2 ISWs (ISW11, ISW12,
ISW16, and ISW17) in the survey line L74. The pycnocline centers corresponding to ISW11, ISW12, ISW16, and ISW17
significantly deviate downwards (see the degree to which the mid-depth of the pycnocline deviates from 1/2 seafloor depth in
Table 2, the positive sign indicates that the pycnocline deviates upward, and the negative sign indicates that the pycnocline
deviates downward). It makes the ISW structure at the top more developed. From ISW11, ISW12, and ISW17 (Fig. 12a, b, d),
it can be seen that the amplitude of the ISW in the upper half of the pycnocline firstly increases and then decreases with the
increasing seawater depth.

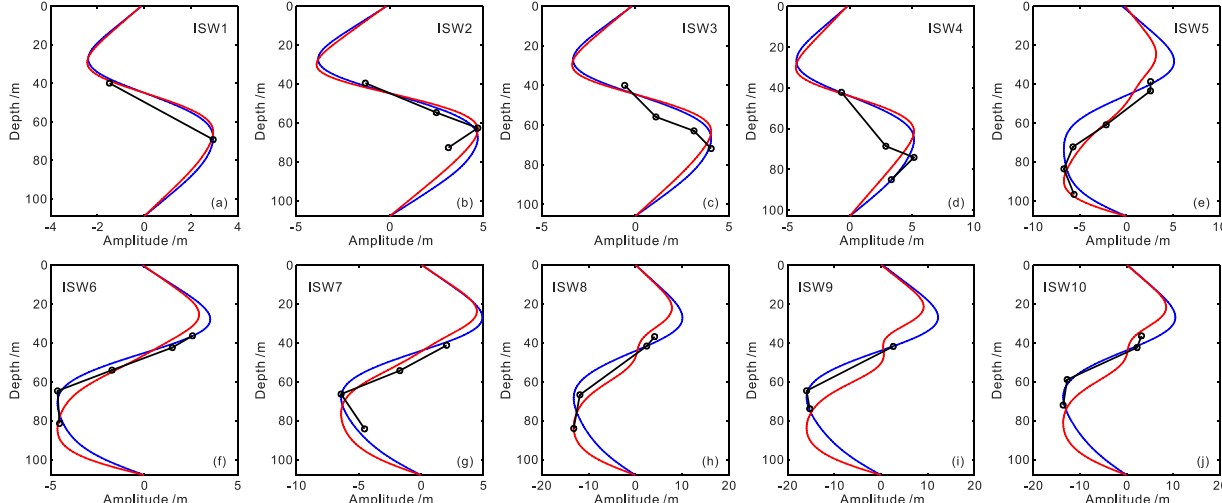


**Figure 11. (a)-(j) respectively demonstrate the vertical structure characteristics of the amplitude of the 10 mode-2 ISWs ISW1-**
**ISW10 in the survey line L84 as well as the vertical mode function fitting results. The black circles denote the observed ISWs'**
**amplitudes at different depths. The blue curves are the linear vertical mode function (nonlinear correction is not considered). And**
**the red curves are the first-order nonlinear vertical mode function (nonlinear correction is considered).**

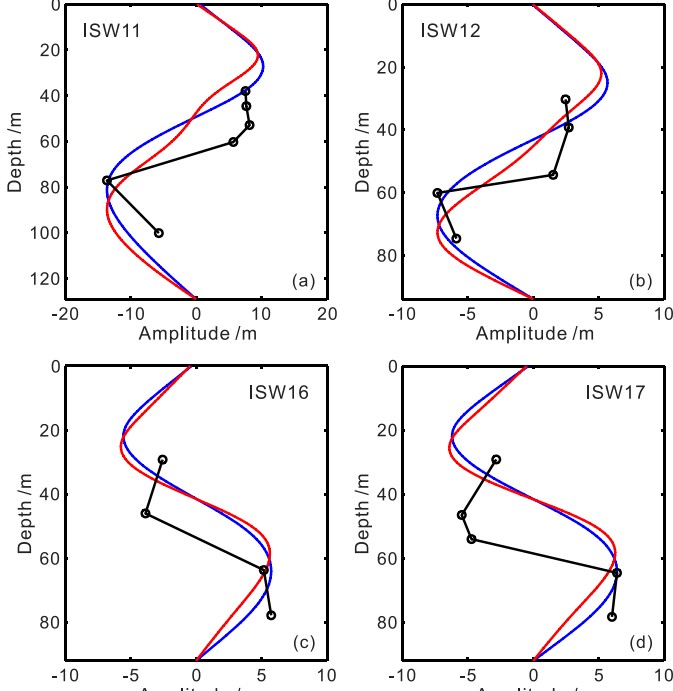


**Figure 12. (a)-(d) respectively demonstrate the vertical structure characteristics of the amplitude of the four mode-2 ISWs (ISW11,**
**ISW12, ISW16, and ISW17) in the survey line L74 as well as the vertical mode function fitting results. The black circles denote the**
**observed ISWs' amplitudes at different depths. The blue curves are the linear vertical mode function (nonlinear correction is not**
**considered). And the red curves are the first-order nonlinear vertical mode function (nonlinear correction is considered).**

To study the vertical structure of the mode-2 ISW amplitude in more detail for the study area, we respectively compare
the observation result with the linear vertical mode function (nonlinear correction is not considered, the blue curves in Figs.
11 and 12) and the first-order nonlinear vertical mode function (considering nonlinear correction, the red curves in Figs. 11
and 12). The linear vertical mode function can be obtained by solving the eigenvalue equation that satisfies the Taylor-
Goldstein problem (Holloway et al., 1999):

$$\frac{d^2\varphi(z)}{dz^2} + \frac{N^2(z)}{C^2}\varphi(z) = 0$$


$$\varphi(0) = \varphi(-H) = 0 \tag{6}$$


where $\varphi(z)$ represents the linear vertical mode function, $C$ is the linear phase speed, $N(z)$ is the Brunt-Väisälä frequency. We
use the temperature and salinity data coming from CMEMS (Copernicus Marine Environment Monitoring Service) to compute
Brunt-Väisälä frequency. The first-order nonlinear vertical mode function is obtained by adding a nonlinear correction term to
the linear vertical mode function (Lamb and Yan, 1996). It can be expressed by the equation as follow:

$$\varphi_m(z) = \varphi(z) + \eta_0 T(z) \tag{7}$$


where $\eta_0$ is the ISW maximum amplitude in the vertical direction, and $T(z)$ is the first-order nonlinear correction term. $T(z)$
satisfies an inhomogeneous equation as follow (Grimshaw et al., 2002, 2004):

$$\frac{d^2T(z)}{dz^2} + \frac{N(z)^2}{C^2}T(z) = -\frac{\alpha}{C}\frac{d^2\varphi(z)}{dz^2} + \frac{3}{2}\frac{d}{dz}\left[\left(\frac{d\varphi(z)}{dz}\right)^2\right] \tag{8}$$


$$T(0) = T(-H) = 0$$


$$\alpha = \frac{3C}{2}\frac{\int_{-H}^{0}\left(\frac{d\varphi(z)}{dz}\right)^3 dz}{\int_{-H}^{0}\left(\frac{d\varphi(z)}{dz}\right)^2 dz} \tag{9}$$


where $\alpha$ is the quadratic nonlinear coefficient. Equation (8) has a unique solution by adding the restriction condition of
$T(z_{max})=0$ (Grimshaw et al., 2002), where $z_{max}$ represents the depth of the maximum amplitude of ISW. The detailed calculation
process is described in Gong et al. (2021). The fitting effects of the linear vertical mode function and the first-order nonlinear
vertical mode function in Figs. 11 are shown in Table 6. We comprehensively evaluate the goodness of fitting by the computed
$R^2$, the depths corresponding to the maximum amplitude between the observation results and the fitting results, and the overall
trends between the observation results and the fitting results. Observing Fig. 11 and Table 6, it can be found that the overall
nonlinearity of the ISWs ISW5 (Fig. 11e) and ISW8 (Fig. 11h) on the survey line L84 is relatively strong. And the first-order
nonlinear vertical mode function considering nonlinear correction can be used to better fit the vertical structure of the amplitude
(the red curves in Fig. 11e, h). The nonlinearity is relatively strong at the bottom of ISW2 (the seawater depth range is 60-80
m in Fig. 11b), the top of ISW7 (the seawater depth range is 40-60 m in Fig. 11g), and the top of ISW10 (the seawater depth
is about 40 m in Fig. 11j). And the first-order nonlinear vertical mode function considering nonlinear correction can be used
to better fit the vertical structure of the amplitude (the red curves in Fig. 11b, g, j). The overall nonlinearity of ISW1 (Fig. 11a),
ISW3 (Fig. 11c), ISW6 (Fig. 11f), and ISW9 (Fig. 11i) is relatively weak. And the linear vertical mode function can be used

to better fit the vertical structure of the amplitude. The nonlinearity is relatively weak at the top of ISW2 (the seawater depth range is 40-60 m in Fig. 11b), the bottom of ISW7 (the seawater depth range is 60-90 m in Fig. 11g), and the bottom of ISW10 (the seawater depth is below 40 m in Fig. 11j). The linear vertical mode function can be used to better fit the vertical structure of the amplitude (the blue curves in Fig. 11b, g, j). The above analysis reflects that the vertical structure of the mode-2 ISW amplitude in the study area is affected by the nonlinearity degree of the ISW. The fitting effects of the linear vertical mode function and the first-order nonlinear vertical mode function in Figs. 12 are shown in Table 7. We comprehensively evaluate the goodness of fitting by the computed $R^2$, the depths corresponding to the maximum amplitude between the observation results and the fitting results, and the overall trends between the observation results and the fitting results. Observing Fig. 12 and Table 7, it can be found that neither the linear vertical mode function (without considering nonlinear correction) nor the first-order nonlinear vertical mode function (with consideration of nonlinear correction) can be used to well fit the vertical structure of the amplitude of the ISWs ISW11, ISW12, ISW16, and ISW17 on L74 (especially the position of the upper half of the pycnocline). The ISWs ISW11, ISW12, ISW16, and ISW17 on the survey line L74 have the large downward deviation of the pycnocline center (see the degree to which the mid-depth of the pycnocline deviates from 1/2 seafloor depth in Table 2, the positive sign indicates that the pycnocline deviates upward, and the negative sign indicates that the pycnocline deviates downward). We have observed the fitting result of the vertical amplitude of the ISW with the large downward pycnocline deviation on other lines of the study area (not shown in this article). And we found that the fitting result of the vertical amplitude is usually poorer than that of the ISW corresponding to the upward deviation of the pycnocline (especially the position of the upper half of the pycnocline). We believe that when the pycnocline center has a large downward deviation, the vertical mode function (including the linear vertical mode function without considering nonlinear correction and the first-order nonlinear vertical mode function considering nonlinear correction) cannot be used to well fit the vertical structure of the mode-2 ISW amplitude in the study area. The above analysis once again reflects that the pycnocline deviation (especially the downward deviation of the pycnocline) affects the vertical structure of the mode-2 ISW amplitude in the study area. In addition, we could not find a good way to fit the vertical amplitude structure in Fig. 12 based on the basic KdV theory. Maybe it needs other theory to fit this kind of vertical amplitude structure. We hope it could be solved in the following study.

**Table 6. The Fitting Effects of Each Curve in Figure 11 on the Observation Points.**

| ISW# | seawater depth range | fitting curve | $R^2$ |
|---|---|---|---|
| ISW1 | 39-70 m | blue curve | 0.98 |
| | | red curve | 0.99 |
| ISW2 | 39-63 m | blue curve | 0.96 |
| | | red curve | 0.88 |
| ISW2 | 63-73 m | blue curve | less than 0 |
| | | red curve | 0.09 |

| ISW# | seawater depth range | fitting curve | $R^2$ |
|---|---|---|---|
| ISW3 | 39-72 m | blue curve | 0.59 |
| | | red curve | 0.4 |
| ISW4 | 42-86 m | blue curve | 0.72 |
| | | red curve | 0.71 |
| ISW5 | 38-97 m | blue curve | 0.81 |
| | | red curve | 0.94 |
| ISW6 | 36-82 m | blue curve | 0.97 |
| | | red curve | 0.94 |
| ISW7 | 41-66 m | blue curve | 0.8 |
| | | red curve | 0.91 |
| ISW7 | 66-85 m | blue curve | 0.77 |
| | | red curve | less than 0 |
| ISW8 | 36-85 m | blue curve | 0.95 |
| | | red curve | 0.95 |
| ISW9 | 41-74 m | blue curve | 1 |
| | | red curve | 0.8 |
| ISW10 | 36-42 m | blue curve | less than 0 |
| | | red curve | less than 0 |
| ISW10 | 42- 73 m | blue curve | 0.99 |
| | | red curve | 0.65 |

514

515 **Table 7. The Fitting Effects of Each Curve in Figure 12 on the Observation Points.**

| ISW# | seawater depth range | fitting curve | $R^2$ |
|---|---|---|---|
| ISW11 | 37-101 m | blue curve | 0.22 |
| | | red curve | 0.3 |
| ISW12 | 30-75 m | blue curve | 0.47 |
| | | red curve | 0.68 |
| ISW16 | 29-78 m | blue curve | 0.46 |
| | | red curve | 0.25 |
| ISW17 | 29-79 m | blue curve | less than 0 |
| | | red curve | less than 0 |

## 4 Discussion

As for the relationship between the dimensionless propagation speed $\tilde{U}$ and the dimensionless amplitude $\tilde{a}$ of the mode-2 ISW in the study area, as well as the relationship between the dimensionless wavelength $\lambda_0$ and the dimensionless amplitude $\tilde{a}$, both of them are not strictly monotonous in the case of smaller amplitude ($\tilde{a}<1$) and show the characteristics of multi-parameter controlling. For this reason, we analyzed the influence of seawater depth on the $\tilde{U}$ and $\lambda_0$ of the mode-2 ISW in the study area. The results are shown in Fig. 13a and b, respectively. Observing Fig. 13a, it can be found that in the shallow seawater (the seafloor depth is less than 120 m) the $\tilde{U}$ variation range is small. And there are both the large-amplitude mode-2 ISWs ($\tilde{a}>2$) and the small-amplitude mode-2 ISWs ($\tilde{a}<2$). In the deep seawater (or at the shelf break, the seafloor depth is greater than 120m), the smaller-amplitude mode-2 ISWs ($\tilde{a}<1$, dark blue filled circles in Fig. 13a) have a large $\tilde{U}$ variation range. The maximum $\tilde{U}$ can reach 2.45, and the minimum can reach 0.5. In particular, the smaller $\tilde{U}$ values are mainly concentrated in the deep seawater, so that in Fig. 7 when $\tilde{a}<1.18$ the relationship between the $\tilde{U}$ and the $\tilde{a}$ of the mode-2 ISW seems to have the trend given by Kozlov and Makarov (1990). The sharp decrease in the $\tilde{U}$ values of the mode-2 ISWs with smaller amplitudes in the deep seawater may be caused by the collision of the ISWs with the seafloor topography (including the step) at the shelf break. In addition, from Fig. 10a and b, it can be found that on the whole, the pycnocline depths and the pycnocline thicknesses of the larger-amplitude mode-2 ISWs ($\tilde{a}>1$) are respectively smaller than the pycnocline depths and the pycnocline thicknesses of the smaller-amplitude mode-2 ISWs ($\tilde{a}<1$). Therefore, the propagation speeds of the larger-amplitude mode-2 ISWs ($\tilde{a}>1$) are generally smaller than the propagation speeds of the smaller-amplitude mode-2 ISWs ($\tilde{a}<1$). In Fig. 7 when $\tilde{a}>1.18$, this makes the relationship between the $\tilde{U}$ and the $\tilde{a}$ of the mode-2 ISW is closer to the result predicted by the deep-water weakly nonlinear theory (Benjamin, 1967). The above-analyzed influences of the seawater depth (seafloor topography), the pycnocline depth, and the pycnocline thickness on the mode-2 ISW propagation speed of the study area, have caused the diversity of the relationship between $\tilde{U}$ and $\tilde{a}$. That is, when $\tilde{a}<1.18$, the relationship between the $\tilde{U}$ values and the $\tilde{a}$ values of the observed mode-2 ISWs in the study area seems to have the trends respectively given by Kozlov and Makarov (1990), as well as Salloum et al. (2012). When $\tilde{a}>1.18$, the relationship between the $\tilde{U}$ values and the $\tilde{a}$ values of the observed mode-2 ISWs in the study area is closer to the result predicted by the deep-water weakly nonlinear theory (Benjamin, 1967).

Observing Fig. 13b, it can be found that the mode-2 ISWs with the smaller amplitudes ($\tilde{a}<1$, the dark blue filled circles in Fig. 13b) have a relatively large variation range of the dimensionless wavelength $\lambda_0$ in the deep seawater (the seafloor depth is greater than 120m). The largest $\lambda_0$ can reach up to 9.55 (corresponding to ISW2 on the survey line L84, whose pycnocline deviation is large and waveform is asymmetric). And the smallest $\lambda_0$ can reach 2.44, so that the $\lambda_0$ of the vertical axis in Fig. 8 can be reduced to 2.44 when $\tilde{a}<1$. The sharp decrease in the $\lambda_0$ values of the mode-2 ISWs with the smaller amplitudes ($\tilde{a}<1$) in deep seawater may be caused by the collision of the ISWs with the seafloor topography at the shelf break. The sharp increase in $\lambda_0$ values of the mode-2 ISWs with the smaller amplitudes in deep seawater may be related to the waveform asymmetry caused by the pycnocline deviation.

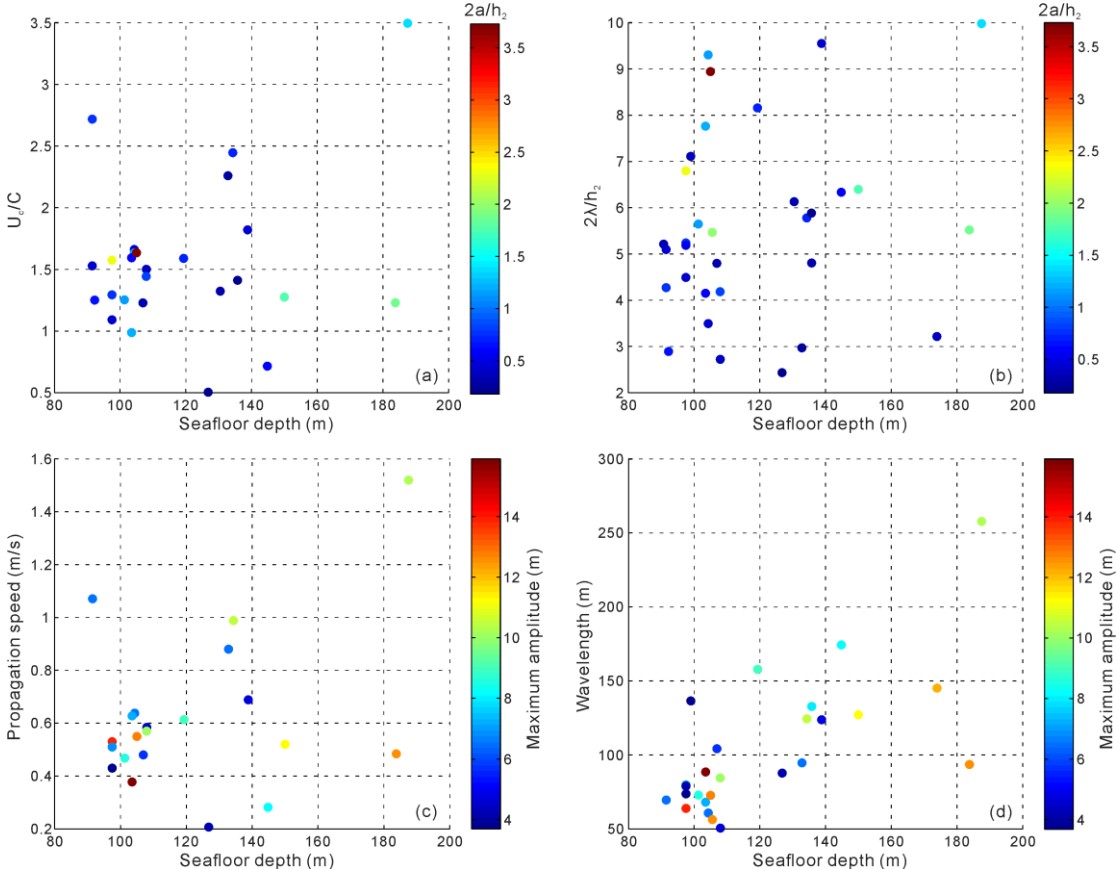

Figure 13. (a) the relationship between the dimensionless propagation speeds and the seawater depths of the mode-2 ISWs observed in the study area. The color of the filled circle indicates the dimensionless amplitude. (b) The relationship between the dimensionless wavelengths and the seawater depths of the mode-2 ISWs observed in the study area. The color of the filled circle indicates the dimensionless amplitude. (c) The relationship between the propagation speeds and the seawater depths of the mode-2 ISWs observed in the study area. The color of the filled circle indicates the maximum amplitude. (d) The relationship between the wavelengths and the seawater depths of the mode-2 ISWs observed in the study area. The color of the filled circle indicates the maximum amplitude.

Fig. 13c and d respectively show the relationship between the propagation speed $U_c$ and the seawater depth, and the relationship between the wavelength $\lambda$ and the seawater depth of the mode-2 ISW in the study area. The color of the filled circles in the figures represents the maximum amplitude. Observing Fig. 13c, we find that the seawater depth in the study area has a great influence on the $U_c$ of the mode-2 ISW. In the shallow seawater area (the seawater depth is less than 120 m), the $U_c$ variety range is small. In the deep seawater area (the seawater depth is larger than 120 m) the $U_c$ has a large variety range. The maximum $U_c$ is 1.52 m/s and the minimum $U_c$ is 0.21 m/s. In Fig. 9a, when the maximum amplitude is between 6 m and 11 m, the $U_c$ has the larger variety range. And there is a significant increase in the $U_c$. The above phenomenon is controlled by the seawater depth. That is, in the deep seawater area (seawater depth greater than 120 m), for the ISWs with the maximum

amplitude of 6-11 m, the $U_c$ varies widely. And the maximum $U_c$ of 1.52 m/s appears (Fig. 13c). Observing Fig. 13d, we find that the seawater depth in the study area has a great influence on the wavelength $\lambda$ of the mode-2 ISW. On the whole, the $\lambda$ of the ISW increases with the increasing seawater depth. For the ISWs with the maximum amplitude of 7-13 m, considerable parts of them are distributed in the deep seawater area (the seawater depth is larger than 120 m), making their $\lambda$ values increase significantly. As a result, when the maximum amplitude is between 7 m and 13 m in Fig. 9b, there is a significant increase in the wavelength $\lambda$.

McSweeney et al. (2020a, 2020b) conducted observational studies on the cross-shore and alongshore evolution characteristics of internal bores near Point Sal, California. They used the quadratic nonlinear coefficient $\alpha$ calculated by KdV theory to characterize the stratification. And they found that when the $\alpha$ calculated from the background density is greater than 0, the waveform of the internal bore becomes steep as the internal bore passes the site. When the $\alpha$ calculated from the background density is less than 0, the waveform of the internal bore becomes rarefied as the internal bore passes the site. Background stratification affects the evolution of internal bores. And the passage of an internal bore will also change the stratification, which in turn affects the evolution of a subsequent internal bore. They found that the change in the $\alpha$ after the internal bore passed is positively correlated with the background $\alpha$. By analogy with the work of McSweeney et al. (2020a, 2020b), we calculated the background quadratic nonlinear coefficient $\alpha$ (corresponding to the stratification before the arrival of the ISW) and the linear phase speed $C$, at the position of the ISWs in the study area by solving Eq. (6) and Eq. (9). Because the theoretical vertical structures calculated based on the KdV theory cannot well fit the observed vertical structures of the ISWs on the survey line L74 (Fig. 12). We are not sure that the KdV theory can well describe the ISWs appearing on the survey line L74. Therefore, we have only calculated the $\alpha$ and $C$ at each ISW position on the survey line L84. The calculation results show in columns 12 and 13 of Table 1, respectively. Observing the calculated $\alpha$ values in Table 1, we find that the $\alpha$ values of ISW1-ISW4 are all less than 0. And the $\alpha$ values of ISW5-ISW10 are all greater than 0. It corresponds well to the waveform characteristics of the ISWs in Fig. 3. That is, for ISW1-ISW4 whose $\alpha$ values are less than 0, their waveforms are relatively rarefied. For ISW5-ISW10 whose $\alpha$ values are greater than 0, their waveforms are relatively steep. It indicates that the background stratification has an influence on the shape of the mode-2 ISWs in the study area. Observing the calculated $C$ values in Table 1, we find that from ISW1 to ISW4, the calculated $C$ values gradually decrease with the decreasing seafloor depths. It is consistent with the observed trend that the propagation speeds $U_c$ of the ISWs (column 11 of Table 1) also gradually decrease with the decreasing seafloor depths. ISW5 is shallower than ISW4. But the calculated $C$ and the observed $U_c$ of ISW5 are both greater than those of ISW4. From ISW5 to ISW10, as the seafloor depths gradually decrease, the calculated $C$ values and the observed ISW $U_c$ values overall show a decreasing trend again. We think the above phenomenon is caused by background stratification. That is, ISW1-ISW4 have a similar background stratification. And ISW5-ISW10 have another similar background stratification. It makes the calculated $C$ values and observed $U_c$ values of SW1-ISW4 decrease with the decreasing of seafloor depths. The calculated $C$ and observed $U_c$ of ISW5 are greater than those of ISW4. On the whole, the calculated $C$ values and the observed $U_c$ values of ISW5-ISW10 decrease with the decreasing of seafloor depths. The above

discussion indicates that the background stratification has an influence on the propagation speeds of the mode-2 ISWs in the
study area.

## 5 Conclusions

We carried out a regional study of the mode-2 ISWs in the Pacific coast of Central America using the seismic reflection
method. Through the analysis of the typical seismic sections L84 and L74, we find that when the degree of downward
pycnocline deviation is large, the influence of pycnocline deviation on the stability of the mode-2 ISW is more complicated
than when the pycnocline deviates upwards. There are mode-2 ISWs with the large degree of downward pycnocline deviation
but with the relatively symmetrical waveform.
The observed relationship between the dimensionless propagation speed $\tilde{U}$ and the dimensionless amplitude $\tilde{a}$ of the
mode-2 ISW in the study area is analyzed. When $\tilde{a}<1.18$, $\tilde{U}$ seems to increase with the increasing $\tilde{a}$, divided into two parts
with different growth rates. When $\tilde{a}>1.18$, $\tilde{U}$ increases with the increasing $\tilde{a}$ at a relatively small growth rate. The observed
relationship between the dimensionless wavelength $\lambda_0$ and the dimensionless amplitude $\tilde{a}$ of the mode-2 ISW in the study area
is also analyzed. When $\tilde{a}<1$, $\lambda_0$ seems to change from 2.5 to 7 for a fixed $\tilde{a}$. When $\tilde{a}>1.87$, $\lambda_0$ increases with the increasing $\tilde{a}$.
As for the relationships between $\tilde{U}$ and $\tilde{a}$, as well as $\lambda_0$ and $\tilde{a}$ of the mode-2 ISW in the study area, both of them show the
characteristics of multi-parameter controlling. The seawater depth (seafloor topography), the pycnocline depth, and the
pycnocline thickness have influences on the mode-2 ISW propagation speed of the study area. It causes the diversity of the
relationship between $\tilde{U}$ and $\tilde{a}$.
The vertical structure of the mode-2 ISW amplitude in the study area is affected by the nonlinearity degree of the ISW.
Part of the mode-2 ISWs with the strong nonlinearity (or the part with strong nonlinearity of the ISWs in the vertical direction)
can use the first-order nonlinear vertical mode function (nonlinear correction is considered) to better fit the vertical structure
of the amplitude. The pycnocline deviation (especially the downward deviation of the pycnocline) affects the vertical structure
of the mode-2 ISW amplitude in the study area. When the pycnocline center has a large downward deviation, the vertical mode
function cannot be used to well fit the vertical structure of the mode-2 ISW amplitude in the study area.
**Code and data availability.** The full seismic data are provided by MGDS (The Marine Geoscience Data System)
(http://www.marine-geo.org/), available for academic research at www.marine-geo.org/tools/search/entry.php?id=EW0412.
The temperature and salinity data comes from CMEMS (Copernicus Marine Environment Monitoring Service)
(http://marine.copernicus.en/ services- portfolio/access-to-products/).
**Author contribution.** The concept of this study was developed by Haibin Song and extended upon by all involved. Wenhao
Fan implemented the study and performed the analysis with guidance from Haibin Song. Yi Gong, Shun Yang and Kun Zhang
collaborated in discussing the results and composing the manuscript.
**Competing interests.** The authors declare that they have no conflict of interest.
**Acknowledgements.** We thank the captain, crew, and science party of R/V Maurice Ewing cruise EW0412 for acquiring the
seismic data. We appreciate MGDS and CMEMS for their supporting data used in this study. This work is supported by the
National Natural Science Foundation of China (Grant Number 41976048) and the National Key R&D Program of China
(2018YFC0310000).

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
