# Peer review of "Regional study of mode-2 internal solitary waves in the Pacific coast of Central America using marine seismic survey data"

_Nonlinear Processes in Geophysics, 2021_

## Author Comment (AC1)

**Response to reviewer #1**

We gratefully thank the editor and all reviewers for their time spent making their constructive remarks and useful suggestions, which has significantly raised the quality of the manuscript and has enabled us to improve the manuscript. Each suggested revision and comment, brought forward by the reviewers, was accurately incorporated and considered. Below the comments of the reviewers are responses point by point and the revisions are indicated.

**RC1**

The study reports on characteristics of mode-2 internal waves in the Pacific coast of Central America using marine seismic survey data. Observations of mode-2 internal waves in the ocean are relatively few. The research may contribute to our understanding of this wave phenomenon. I have some major concerns.

**1. Comment:** Line 27: Satellite remote sensing can not see the ocean interior. Thus, there is not an issue of vertical resolution.

**1. Reply:** Thank you for your valuable comment. Our description of the vertical resolution of remote sensing is indeed inappropriate. We have changed our statements in the corresponding places in the introduction of the revised manuscript (lines 25-27). We also show the revised text as follow:

Conventional physical oceanography observation and remote sensing observation have their limitations. That is, the horizontal resolution of conventional physical oceanography observation methods (such as mooring) is low. And satellite remote sensing cannot see the ocean interior.

**2. Comment:** Line 35: Most of the cited references in the paragraph use very ideal stratification. I am not sure how much these researches are relevant to mode-2 internal waves in the ocean.

**2. Reply:** Thank you for your valuable comment. Yes, these researches maybe don't have much relevance to mode-2 internal waves in the ocean because of the use of the very ideal stratification. We feel very sorry for our inaccurate statement and have changed our statement in the corresponding place in the introduction of the revised manuscript (line 36). We also show the revised text as follow:

At present, the researches on the mode-2 ISW are mainly based on simulation.

3. Comment: Line 103: The mode-2 ISWs in the actual ocean has continuous structure?

**3. Reply:** Thank you for your comment. Yes, we think the mode-2 ISWs in the actual ocean have continuous structure based on our observation. That is, the mode-2 ISWs in the actual ocean have multiple continuous density displacements above and below the mid-depth of the pycnocline. We have added the above description in the corresponding place in section 2 (Data and Methods) of the revised manuscript (lines 115-116). We also show the revised text as follow:

But the mode-2 ISW in the actual ocean has a multilayer structure (multiple continuous density displacements above and below the mid-depth of the pycnocline).

**4. Comment:** Line 106: I am sorry I do not understand why the equivalent three-layer model is used to define the amplitude of mode-2 ISWs. In oceanography, the amplitude is defined as the maximum vertical displacement of isopycnals (e.g. Shroyer,2010,JGR).

4. Reply: Thank you for your comment. We noticed that the amplitude, defined as the maximum vertical displacement of isopycnals, is used less in quantitatively describing the amplitude-related characteristics of mode-2 ISW. Particularly, in mode-2 ISW simulation research, the scholars often use the dimensionless amplitude  $2a/h_2$  to quantitatively describe the amplitude-related characteristics of mode-2 ISW, like the relationship between the propagation speed and the dimensionless amplitude (Brandt et al., 2014; Carr et al., 2015). It is important to point that in mode-2 ISW simulation research, the dimensionless amplitude the scholars used comes from the three-layer model, which is different from the continuous structure (multilayer structure) of mode-2 ISW in the actual ocean. Because there is almost no work of the previous scholars to define the dimensionless amplitude of the mode-2 ISW based on the mode-2 ISW in the actual ocean (with multiple continuous density displacements above and below the mid-depth of the pycnocline) for our reference. To compare our observation results to the simulation results and quantitatively describe the amplitude-related characteristics of mode-2 ISW, we try our best to build an equivalent three-layer model. The equivalent three-layer model results from the mode-2 ISW with the continuous structure in the actual ocean. And we use this equivalent three-layer model to define the amplitude (dimensionless amplitude). Besides, we also try to use the maximum amplitude (the maximum vertical displacement of isopycnals) to study the amplitude-related characteristics of mode-2 ISW, like the relationship between the propagation speed and the maximum amplitude in Figure 9. But the correlativity is not very strong. That is another reason we try our best to build the equivalent three-layer model to define the amplitude of mode-2 ISW. We have added the above descriptions in the corresponding places in section 2 (Data and Methods) of the revised manuscript (lines 108-123).

**5. Comment:** Line 135: I am not familiar with the seismic reflection method, and I can not ensure the correctness of Eq.(1) and Eq.(2). However, my intuitive idea is that the actual wave form need to be obtained first.

**5. Reply:** Thank you for your comment. In a seismic survey, the sound is sent from a towed source, reflected from aquatic structures, and received by an array of towed hydrophones with time delays that depends on the geometry of the ray paths taken. The detailed introduction to seismic principles is described by Ruddick et al. (2009). Traditional seismic reflection imaging assumes that the underground structure is fixed. In the seawater, the mode-2 ISWs move relatively fast in the horizontal direction (about 0.5m/s), so the seismic reflection imaging of the mode-2 ISWs needs to consider the influence of the horizontal motion of the ISWs. We believe Eq.(2) and Eq.(3) in the revised manuscript are correct. We have added the above descriptions in the corresponding places in section 2 (Data and Methods) of the revised manuscript (lines 144-149).

**6. Comment:** Line 172: Is the dimensionless amplitude 2a/h2 equivalent to the one used by Brandt et al. 2014? The equivalent three layer model differs from Fig. 1 in Brandt et al. (2014).

**6. Reply:** Thank you for your comment. The dimensionless amplitude 2a/h2 is not completely equivalent to the one used by Brandt et al. 2014. And the equivalent three-layer model is not completely the same as Fig. 1 in Brandt et al. (2014). Because there is almost no work of the previous scholars to define the dimensionless amplitude of the mode-2 ISW based on the mode-2 ISW in the actual ocean (with multiple continuous density displacements above and below the mid-depth of the pycnocline) for our reference. The equivalent three-layer model is defined by trying our best to analogize with the three-layer model. We have added the above descriptions in the corresponding places in section 2 (Data and Methods) of the revised manuscript (lines 123-128). We have also modified the description related to the content of the equivalent amplitude is defined by trying our best to analogize with the three-layer model, and is not completely equivalent to the definition of the three-layer model in the simulation experiment. We show the revised text (lines 196-197) as follow:

We define the ISW, whose  $\tilde{a}$  value (dimensionless amplitude) is less than 2, as the mode-2 ISW with a small amplitude. And define the ISW, whose  $\tilde{a}$  value is larger than 2, as the mode-2 ISW with a large amplitude.

**7. Comment:** Line 322: In Figure 8, the nondimensional wavelength does not seem to decrease with increasing nondimensional amplitude when 2a/h2<1. My observation is that the nondimensional wavelength may change from 2.5 to 7 for a fixed nondimensional amplitude.

**7. Reply:** Thank you for your valuable comment. We realized that it is not accurate to describe the nondimensional wavelength decreases with the increasing nondimensional amplitude when 2a/h2<1. We have changed our statement into "The nondimensional wavelengths seem to change from 2.5 to 7 for a fixed nondimensional amplitude when 2a/h2<1" in the corresponding places in the abstract, section 3.2, and conclusions of the revised manuscript (lines 14-15, lines 331-333, and line 612). We also show the revised text (lines 331-333) as follow:

Observing Fig. 8, it can be found that when  $\tilde{a} < 1$ , the relationship between the  $\lambda_0$  values and the  $\tilde{a}$  values of the observed mode-2 ISWs in the study area is closer to the result predicted by the deep-water weakly nonlinear theory (Benjamin, 1967). But the  $\lambda_0$  values change from 2.5 to 7 for a fixed  $\tilde{a}$  value.

**8. Comment:** Line 400: How is the wave frequency defined? It is very important because the eigenfunction crucially depends on the wave frequency. Moreover, I note that Holloway et al.(1999) do not use wave frequency in the eigenvalue problem.

8. **Reply:** Thank you for your valuable comment. The formula for calculating the wave frequency is  $w=v/\lambda$ , where the wavelength  $\lambda$  (the wavelength here is two times the wavelength used in our manuscript) is measured from stacked seismic section, and the propagation speed v is estimated by the method described in section 2 (Data and Methods) of our manuscript. But actually, we used the w=0 in our study. We try to use the wave frequency to compute the eigenfunction during the revising process. Maybe because the wave frequencies of the mode-2 ISWs we studied are around  $0.003s^{-1}$ , the eigenfunction does not change much when using the wave frequency. In order not to cause misunderstanding, we have deleted the wave

frequency in the eigenvalue equation and the related description in the revised manuscript (lines 461-465). We also show the revised text (lines 461-465) as follow:

The linear vertical mode function can be obtained by solving the eigenvalue equation that satisfies the Taylor-Goldstein problem (Holloway et al., 1999):

$$\frac{d^2\varphi(z)}{dz^2} + \frac{N^2(z)}{C^2}\varphi(z) = 0$$

$$\varphi(0) = \varphi(-H) = 0$$
(6)

where  $\varphi(z)$  represents the linear vertical mode function, *C* is the linear phase speed, *N*(*z*) is the Brunt-Väisälä frequency.

9. Comment: Please consider to reduce the use of long sentences in the manuscript.

**9. Reply:** Thanks for your valuable suggestion. We have reduced the use of long sentences in the manuscript.

---

## Author Comment (AC2)

**Response to reviewer #2**

We gratefully thank the editor and all reviewers for their time spent making their constructive remarks and useful suggestions, which has significantly raised the quality of the manuscript and has enabled us to improve the manuscript. Each suggested revision and comment, brought forward by the reviewers, was accurately incorporated and considered. Below the comments of the reviewers are responses point by point and the revisions are indicated.

**RC2**

The manuscript seeks to quantify the relationship between the propagation speed and wavelength of mode-2 ISWs and a number of parameters (e.g. depth). Data is from measurements taken during a field campaign off the coast of Central America. The manuscript is difficult to read and its scientific conclusions are weak. I have worked on mode-2 waves for some time, and the question of propagation speed/wavelength variability never struck me as a pressing issue. This doesn't mean I wouldn't want to read about it, but it does mean I would like some motivation for doing so. The authors hint at links with actual dynamics of the mode-2 waves (lines 30-50) but never really return to these ideas when discussing their results.

**Response:** Thank you for your comment. Because generally, the seismic oceanography method can only get the "snapshot" of ocean vertical temperature gradient for a long horizontal distance section. It is a little difficult to study the actual dynamics of the mode-2 ISWs during a long period using the seismic oceanography method. The kinematics characteristics of the ISWs we want to describe in lines 30-50 are mainly the characteristics related to the wave propagation speed. We have added the supplementary explanation in the corresponding places in the introduction of the revised manuscript (line 33). We also show the revised text as follow:
Scholars have used the seismic oceanography method to carry out related studies on the geometry and kinematics characteristics (mainly related to propagation speed) of ISW in the South China Sea, the Mediterranean Sea, and the Pacific Coast of Central America (Bai et al., 2017; Fan et al., 2021a, 2021b; Geng et al., 2019; Sun et al., 2019; Tang et al., 2014, 2018).

The authors appear to have some rather basic points of theoretical misunderstanding; for example there is repeated mention of phase velocity, but ISWs do not have a phase velocity. Only linear waves have a well-defined distinction between phase and group speeds, something that goes back (at least) to the classical work of Whitham in the 1960s. In a similar vein, ISWs are "long waves" so why does the frequency appear in the TG equation on line 400?

**Response:** Thank you for your valuable comment. We feel sorry for our theoretical misunderstanding. We have changed all the "phase velocity" into "propagation speed" in the revised manuscript. We used the frequency $w=0$ in our study when computing the linear vertical mode function by solving TG equation. It is our mistake to add the frequency in the TG equation. We have deleted the frequency in the TG equation in the corresponding place in the revised manuscript (line 463). We also show the revised text as follow:
The linear vertical mode function can be obtained by solving the eigenvalue equation that

satisfies the Taylor-Goldstein problem (Holloway et al., 1999):

$$\frac{d^2\varphi(z)}{dz^2} + \frac{N^2(z)}{C^2}\varphi(z) = 0$$
$$\varphi(0) = \varphi(-H) = 0 \tag{6}$$

where $\varphi(z)$ represents the linear vertical mode function, $C$ is the linear phase speed, $N(z)$ is the Brunt-Väisälä frequency.

The technical English of the manuscript is quite poor (there are more issues with plural vs singular and verb tense than I could count), to the point of interfering with the scientific messaging. I suppose this could be cleaned up during the peer review process, but the fact that equations are not properly type set, making the quantitative aspects very difficult to judge, suggests not enough care was placed on communication and that this goes beyond language issues. As a simple example, in equation (1) were the fraction typeset as a fraction, the symbol ∗ would be unnecessary and potential confusion with the use of ∗ to denote convolution could be avoided.

**Response:** Thank you for your valuable comment. We feel very sorry for our poor technical English and the equations are not properly typeset. We have tried our best to correct the issues with plural vs singular and verb tense in the revised manuscript. And we have also tried our best to properly typeset the equations in the revised manuscript. In equation (2) of the revised manuscript, we have deleted the symbol "∗". The revised equation is shown as follow:

$$\lambda = \lambda_s - x_w = \lambda_s - \frac{\lambda_s}{V_{ship}}V_{water} \tag{2.}$$

The technical graphics (13 figures) are quite good, with nicely put together Matlab figures and a very nice diagram in Figure 2. Some figures could use grid lines, and Fig 9 could use a better symbol (a filled triangle, '^', or pentagram, 'p' with 'MarkerFaceColor' set in the plot command). I was really surprised that no plots of the density profile the waves propagate on were provided (even in schematic form).

**Response:** Thank you for your valuable suggestion and comment. We have used gridlines in Fig 9 and Fig 10 of the revised manuscript. And a better symbol (a filled triangle) has been used in Fig 9 of the revised manuscript. Previous scholars demonstrated that seismic reflections generally track isopycnal surfaces (Holbrok et al., 2013; Krahmann et al., 2009; Sheen et al., 2011). We believe the seismic stacked sections (like Fig 3 and Fig 4) include the information of the density profile. So we did not provide the plots of the density profile the waves propagate on (even in schematic form). We have added the above description in section 2 (Data and Methods) of the revised manuscript (lines 104-107).

[Figure]

**Figure 9. (a) The relationship between the propagation speeds and the maximum amplitudes of the mode-2 ISWs observed in the study area. (b) The relationship between the wavelengths and the maximum amplitudes of the mode-2 ISWs observed in the study area. The black and red crosses denote the ISWs on the survey lines in the SW-NE direction and in the NE-SW direction, respectively.**

[Figure]

**Figure 10. (a) The relationship between the propagation speeds and the pycnocline depths of the mode-2 ISWs observed in the study area. (b) The relationship between the propagation speeds and the pycnocline thicknesses of the mode-2 ISWs observed in the study area. The color filled circle indicates the dimensionless amplitude.**

Many of the fits shown are very poor (Fig 7,8,10) to the point where there really isn't much one can conclude scientifically (e.g. saying that the red crosses in Figs. 7,8 follow Kozlov and Makarov, as the authors do is rather dubious). For Figure 12, what would a "good fit" be? The procedure for how the red curve in this figure is computed is not clearly described.

**Response:** Thank you for your valuable comment. We acknowledge that many of the fits shown are not very good. We think it is the result of the multi-parameter controlling in the actual ocean. We have changed our descriptions of the fitting results in the revised manuscript to avoid being too absolute. We could not find a "good fit" for Figure 12 based on the basic KdV theory. Maybe it needs other theory to fit this kind of vertical amplitude structure. We hope it could be solved in the following study. The above description has been added in

section 3.3 of the revised manuscript (lines 509-511). We have also added a more detailed description of how the red curves (considering first-order nonlinear correction) in Figures 11 and 12 are computed in section 3.3 of the revised manuscript (lines 470-477). We show the revised text (lines 470-477) as follow:

$T(z)$ is the first-order nonlinear correction term. $T(z)$ satisfies an inhomogeneous equation as follow (Grimshaw et al., 2002, 2004):

$$\frac{d^2T(z)}{dz^2} + \frac{N(z)^2}{C^2}T(z) = -\frac{\alpha}{C}\frac{d^2\varphi(z)}{dz^2} + \frac{3}{2}\frac{d}{dz}\left[\left(\frac{d\varphi(z)}{dz}\right)^2\right] \tag{8}$$

$$T(0) = T(-H) = 0$$

$$\alpha = \frac{3C}{2}\frac{\int_{-H}^{0}\left(\frac{d\varphi(z)}{dz}\right)^3 dz}{\int_{-H}^{0}\left(\frac{d\varphi(z)}{dz}\right)^2 dz} \tag{9}$$

where $\alpha$ is the quadratic nonlinear coefficient. Equation (8) has a unique solution by adding the restriction condition of $T(z_{max})=0$ (Grimshaw et al., 2002), where $z_{max}$ represents the depth of the maximum amplitude of ISW. The detailed calculation process is described in Gong et al. (2021).

I don't want to pile on, and I feel the methodology is quite novel and hence deserves its place in the literature, but the theoretical errors must be fixed before this work is published. So let me lay out what I feel the bottom line is:

**1. Comment:** The theoretical side of this manuscript needs to be cleaned up. The term "phase velocity" needs to be removed, or the authors have to define what it means for ISWs. Equations must be properly typeset, and not just thrown into text as ratios (you could for example define commonly used ratios (e.g. $\tilde{a}=2a/h_2$ and then refer to $\tilde{a}$ in the text).

**1. Reply:** Thank you for your valuable comment and suggestion. We have changed all the "phase velocity" into "propagation speed" in the revised manuscript. We have also tried our best to properly typeset the equations in the revised manuscript. In equation (2) of the revised manuscript, we have deleted the symbol "⋆", and the revised equation (2) are shown as follow:

$$\lambda = \lambda_s - x_w = \lambda_s - \frac{\lambda_s}{V_{ship}}V_{water} \tag{2}.$$

We have defined commonly used ratios to avoid throwing them into text in the revised manuscript (we define $\tilde{a}=2a/h_2$, $\lambda_0=2\lambda/h_2$, $\tilde{U}=U_c/C$, and then refer to $\tilde{a}$, $\lambda_0$, $\tilde{U}$ in the text).

**2. Comment:** Fits need to have R values computed (Matlab will do this via the "tools" button on the figure, or from the command line) and these need to be reported in tables.

**2. Reply:** Thank you for your valuable comment and suggestion. We have computed the $R^2$ values for the fits in Figures 7, 8, 10, 11, and 12. And we have respectively reported them in Tables 3, 4, 5, 6, and 7 of the revised manuscript. We have also added the related description in the text of the revised manuscript (like lines 315-318, lines 335-338, lines 477-480). We show the revised text (lines 315-318, lines 335-338, lines 477-480) as follow:

[revised manuscript text omitted]

**3. Comment:** Give some idea as to the vertical structure of the density field. The readers should understand why the authors want to provide the fits that they do, and they need to have a context for how the density may be changing due to various environmental factors.

**3. Reply:** Thank you for your valuable comment. We have added these contents in the corresponding place in section 3.3 of the revised manuscript (lines 390-415). We also show the revised text (lines 390-415) as follow:

The vertical distribution of ISW amplitude (the vertical displacement of isopycnal) is called its vertical structure. ISWs have different modes, which correspond to different vertical structures (Fliegel and Hunkins, 1975). Previous scholars have used different theoretical models to study the vertical structure of ISW amplitude (Fliegel and Hunkins, 1975; Vlasenko et al., 2000; Small and Hornby, 2005). Among them, only Vlasenko et al. (2000) compared the results of numerical simulation with the results of local observations. And they found that the depths correspond to the ISW maximum amplitude (the maximum vertical displacement of isopycnals) given by the two are in good agreement with each other. At present, there is less work comparing the theoretical vertical structure of mode-2 ISW amplitude with the observed results. This work is conducive to improving our understanding of the vertical structure of the mode-2 ISW in the ocean (including the factors that affect the vertical structure). It can also test the validity and applicability of the theoretical vertical structure to a certain extent. The seismic oceanographic method has a high spatial resolution, and its clear imaging results of ISWs are more conducive to the study of vertical structure. The vertical structure of ISW amplitude is controlled by a variety of environmental factors. Geng et al. (2019) used the seismic oceanography method to study the vertical structure of ISW amplitude near Dongsha Atoll in the South China Sea. It is found that when the ISW interacts intensely with the seafloor, the observed vertical structure of ISW amplitude may be significantly different from the theoretical result. Gong et al. (2021) compared the vertical structure of ISW estimated by theoretical models with the vertical structure of ISW observed by the seismic oceanography method. And they analyzed in detail the factors affecting the vertical structure of ISW amplitude near Dongsha Atoll in the South China Sea. It is found that the vertical structure of ISW is mainly controlled by nonlinearity. It usually appears that the quadratic nonlinear coefficients of ISWs that conform to the linear vertical structure function are small, while the quadratic nonlinear coefficients of ISWs conforming to the first-order nonlinear vertical structure function are larger. In addition, topography, ISW amplitude, seawater depth, and background flow may all affect the vertical structure of ISW amplitude. It appears that larger seawater depth may weaken the influence of the nonlinearity of the ISW on the vertical structure, making the vertical structure of ISW more in line with linear theory. Larger amplitude will make ISW more susceptible to the influence of topography, which will change the vertical structure. Vlasenko et al. (2000) observed that the vertical structure of ISW has local extrema. They thought it is caused by smaller-scale internal waves. In addition, the background flow shear also has an important effect on the vertical structure (Stastna et al., 2002; Liao et al., 2014). Xu et al. (2020) found that the background flow at the center of the eddy can weaken the amplitude of ISW.

**4. Comment:** I would strongly suggest having a look at the various JPO papers on the Pt Sal

internal wave "mega project"; the two McSweeney et al papers especially. Some of these papers included using KdV theory to examine environmental variability in a way that may be useful to the authors, others adopted different points of view which the authors may find useful to put their results in context.

**4. Reply:** Thank you for your valuable suggestion. We have added these contents in Table 1 and the corresponding place in section 4 of the revised manuscript (lines 572-600). We also show the revised text (lines 572-600) as follow:

McSweeney et al. (2020a, 2020b) conducted observational studies on the cross-shore and alongshore evolution characteristics of internal bores near Point Sal, California. They used the quadratic nonlinear coefficient $\alpha$ calculated by KdV theory to characterize the stratification. And they found that when the $\alpha$ calculated from the background density is greater than 0, the waveform of the internal bore becomes steep as the internal bore passes the site. When the $\alpha$ calculated from the background density is less than 0, the waveform of the internal bore becomes rarefied as the internal bore passes the site. Background stratification affects the evolution of internal bores. And the passage of an internal bore will also change the stratification, which in turn affects the evolution of a subsequent internal bore. They found that the change in the $\alpha$ after the internal bore passed is positively correlated with the background $\alpha$. By analogy with the work of McSweeney et al. (2020a, 2020b), we calculated the background quadratic nonlinear coefficient $\alpha$ (corresponding to the stratification before the arrival of the ISW) and the linear phase speed $C$, at the position of the ISWs in the study area by solving Eq. (6) and Eq. (9). Because the theoretical vertical structures calculated based on the KdV theory cannot well fit the observed vertical structures of the ISWs on the survey line L74 (Fig. 12). We are not sure that the KdV theory can well describe the ISWs appearing on the survey line L74. Therefore, we have only calculated the $\alpha$ and $C$ at each ISW position on the survey line L84. The calculation results show in columns 12 and 13 of Table 1, respectively. Observing the calculated $\alpha$ values in Table 1, we find that the $\alpha$ values of ISW1-ISW4 are all less than 0. And the $\alpha$ values of ISW5-ISW10 are all greater than 0. It corresponds well to the waveform characteristics of the ISWs in Fig. 3. That is, for ISW1-ISW4 whose $\alpha$ values are less than 0, their waveforms are relatively rarefied. For ISW5-ISW10 whose $\alpha$ values are greater than 0, their waveforms are relatively steep. It indicates that the background stratification has an influence on the shape of the mode-2 ISWs in the study area. Observing the calculated $C$ values in Table 1, we find that from ISW1 to ISW4, the calculated $C$ values gradually decrease with the decreasing seafloor depths. It is consistent with the observed trend that the propagation speeds $U_c$ of the ISWs (column 11 of Table 1) also gradually decrease with the decreasing seafloor depths. ISW5 is shallower than ISW4. But the calculated $C$ and the observed $U_c$ of ISW5 are both greater than those of ISW4. From ISW5 to ISW10, as the seafloor depths gradually decrease, the calculated $C$ values and the observed ISW $U_c$ values overall show a decreasing trend again. We think the above phenomenon is caused by background stratification. That is, ISW1-ISW4 have a similar background stratification. And ISW5-ISW10 have another similar background stratification. It makes the calculated $C$ values and observed $U_c$ values of SW1-ISW4 decrease with the decreasing of seafloor depths. The calculated $C$ and observed $U_c$ of ISW5 are greater than those of ISW4. On the whole, the calculated $C$ values and the observed $U_c$ values of ISW5-ISW10 decrease with the decreasing of seafloor depths. The above discussion indicates that the background stratification has an

influence on the propagation speeds of the mode-2 ISWs in the study area.

**Table 1. Characteristic Parameters of the 10 Mode-2 Internal Solitary Waves in Survey Line L84.**

| ISW# | $H$ (m) | $A$ (m) | $a$ (m) | $h_2$ (m) | $2a/h_2$ | $\cdots$ | $2\lambda/h_2$ | $U_c$ (m/s) | $\alpha$ ($s^{-1}$) | $C$ (m/s) |
|------|------|------|------|------|------|------|------|------|------|------|
| ISW1 | 145.5 | 3 | 2.22 | 29.23 | 0.15 | $\cdots$ | 7.09 | 0.85±0.6 | -0.018 | 0.384 |
| ISW2 | 138.8 | 4.7 | 5.84 | 25.93 | 0.45 | $\cdots$ | 9.55 | 0.69±0.19 | -0.0179 | 0.382 |
| ISW3 | 130.5 | 4.1 | 4.45 | 27.6 | 0.32 | $\cdots$ | 6.13 | 0.52±0.12 | -0.0181 | 0.378 |
| ISW4 | 121.5 | 5.2 | 6.04 | 34.72 | 0.35 | $\cdots$ | 3.18 | 0.19±0.11 | -0.018 | 0.372 |
| ISW5 | 111 | 6.79 | 12.67 | 40.84 | 0.62 | $\cdots$ | 4.67 | 0.32±0.16 | 0.0068 | 0.391 |
| ISW6 | 108 | 4.6 | 7.5 | 37.19 | 0.4 | $\cdots$ | 2.72 | 0.58±0.16 | 0.0108 | 0.389 |
| ISW7 | 104.3 | 6.4 | 7.34 | 34.83 | 0.42 | $\cdots$ | 3.49 | 0.64±0.28 | 0.0158 | 0.386 |
| ISW8 | 103.5 | 13.2 | 15.82 | 32.94 | 0.96 | $\cdots$ | 4.43 | 0.46±0.24 | 0.0155 | 0.385 |
| ISW9 | 103.5 | 15.9 | 13.56 | 22.79 | 1.19 | $\cdots$ | 7.76 | 0.38±0.17 | 0.0161 | 0.385 |
| ISW10 | 102.8 | 13.6 | 15.87 | 20.62 | 1.54 | $\cdots$ | 9.13 | 0.55±0.34 | 0.0164 | 0.384 |

**Note. $H$, seafloor depths. $A$, maximum amplitudes. $a$, equivalent ISW amplitudes. $h_2$, equivalent pycnocline thicknesses. $\lambda$, equivalent wavelengths. $U_c$, apparent propagation speeds obtained from seismic observation. $\alpha$, quadratic nonlinear coefficient shown in Ea. (9) and is obtained by solving Eq. (6). $C$, linear phase speed which is obtained by solving Eq. (6).**

McSweeney, J. M., Lerczak, J. A., Barth, J. A., Becherer, J., Colosi, J. A., MacKinnon, J. A., MacMahan, J. H., Moum, J. N., Pierce, S. D., and Waterhouse, A. F.: Observations of shoaling nonlinear internal bores across the central California inner shelf, Journal of Physical Oceanography, 50, 111-132, https://doi.org/10.1175/JPO-D-19-0125.1, 2020a.

McSweeney, J. M., Lerczak, J. A., Barth, J. A., Becherer, J., MacKinnon, J. A., Waterhouse, A. F., Colosi, J. A., MacMahan, J. H., Feddersen, F., Calantoni, J., Simpson, A., Celona, S., Haller, M. C., and Terrill, E.: Alongshore variability of shoaling internal bores on the inner shelf, Journal of Physical Oceanography, 50, 2965-2981, https://doi.org/10.1175/JPO-D-20-0090.1, 2020b.

**5. Comment:** Ask a colleague to read strictly for language. Even those like myself, for whom English isn't their first language, could catch simple mistakes (e.g. tense).

**5. Reply:** Thank you for your valuable suggestion. We have asked a colleague to read strictly for language and corrected the grammar mistakes of the manuscript.